# FKBP51 in glutamatergic forebrain neurons promotes early life stress inoculation in female mice

Lotte van Doeselaar[1,2], Alexandra Abromeit [1], Tibor Stark [3,4], Danusa Menegaz[5], Markus Ballmann[6], Shiladitya Mitra[1], Huanqing Yang[1], Ghalia Rehawi [7], Rosa-Eva Huettl [8], Joeri Bordes [1], Sowmya Narayan[1,2], Daniela Harbich[1], Jan M. Deussing [9], Gerhard Rammes[6], Michael Czisch[3], Janine Knauer-Arloth [7,10], Matthias Eder[5], Juan Pablo Lopez [11] & Mathias V. Schmidt [1] ✉

Early life stress (ELS) can increase vulnerability to psychiatric disorders, but also trigger resilience. FKBP51 has been associated with an increased risk for developing psychiatric disorders, specifically in interaction with ELS exposure. Here, the contribution of FKBP51 in glutamatergic forebrain neurons to the long-term consequences of ELS was investigated in both sexes. In female wild-type *Fkbp5^{lox/lox}* mice, ELS exposure led to an anxiolytic phenotype and improved memory performance in a stressful context, however this ELS effect was absent in *Fkbp5^{Nex}* mice. These interactive FKBP51 x ELS effects in female mice were also reflected in reduced brain region volumes, and on structural and electrophysiological properties of CA1 pyramidal neurons of the dorsal hippocampus. In contrast, the behavioral, structural and functional effects in male ELS mice were less pronounced and independent of FKBP51. RNA sequencing of the hippocampus revealed the transcription factor 4 (TCF4) as a potential regulator of the female interactive effects. Cre-dependent viral overexpression of TCF4 in female Nex-Cre mice led to similar beneficial effects on behavior as the ELS exposure. This study demonstrates a sex-specific role for FKBP51 in mediating the adaptive effects of ELS on emotional regulation, cognition, and neuronal function, implicating TCF4 as a downstream effector.

Patients with psychiatric disorders, such as major depressive disorder (MDD) or anxiety disorders, suffer from mood-related and cognitive symptoms, often preventing them from fully participating in society. As a consequence, psychiatric disorders are an important and costly global health problem, for which the biological underpinnings are still poorly understood. In the past decades, it has become clear that psychiatric disease often arises as a combination of genetic and environmental factors[1,2]. Environmental stress exposure can occur at any stage

[1]Research Group Neurobiology of Stress Resilience, Max Planck Institute of Psychiatry, Munich, Germany. [2]International Max Planck Research School for Translational Psychiatry, Munich, Germany. [3]Core Unit Neuroimaging, Max Planck Institute of Psychiatry, Munich, Germany. [4]Emotion Research Department, Max Planck Institute of Psychiatry, Munich, Germany. [5]Core Unit Electrophysiology, Max Planck Institute of Psychiatry, Munich, Germany. [6]Klinik für Anaesthesiologie und Intensivmedizin der Technischen Universität München, Klinikum Rechts der Isar, Munich, Germany. [7]Department Genes & Environment, Max Planck Institute of Psychiatry, Munich, Germany. [8]Core Unit Virus Production, Max Planck Institute of Psychiatry, Munich, Germany. [9]Research Group Molecular Genetics, Max Planck Institute of Psychiatry, Munich, Germany. [10]Computational Health Center, Helmholtz Munich, Neuherberg, Germany. [11]Department of Neuroscience, Karolinska Institute, Stockholm, Sweden. ✉e-mail: mschmidt@psych.mpg.de

in an individual's life and not only the duration, type or severity, but also the timing of stress exposure can be determining for the long-term health outcomes. A wide number of studies have uncovered a time window during early development in which the brain is particularly sensitive to environmental challenges[3–6]. Interestingly, stress exposure during early life can not only lead to maladaptive outcomes but there has also been evidence in rodents, primates and humans for the so-called "match-mismatch" or "inoculation stress" theory[7–10] that proposes that moderate exposure to stress in early life may prepare an individual to cope with future challenges in adulthood. This process of "early programming" may, therefore, act as a long-lasting adaptive mechanism[11,12].

Generally, careful regulation of the stress response is required for adequate stress coping[13], and the hypothalamic pituitary adrenal (HPA) axis is important for keeping this balanced response. Upon perceiving a stressor, the endocrine cascade of the HPA axis leads to the production of cortisol in humans or corticosterone (CORT) in rodents. CORT binds its two receptors that are located in the periphery and brain: the mineralocorticoid receptor (MR) and the glucocorticoid receptor (GR), of which the latter is particularly important in dampening the acute response to stress[13]. As nuclear receptors, MR and GR can influence the transcription of a wide number of genes by binding to glucocorticoid-responsive elements (GREs) that are present on the DNA of numerous genes. One gene that is under strict regulation by GR is the *Fkbp5* gene, encoding the chaperone protein FKBP51. FKBP51, on the other hand, can influence GR sensitivity by binding to the GR complex, thereby hampering GR's transcriptional activity, which results in an ultra-short feedback loop[14,15]. One particular brain region of interest in this regard is the hippocampus. The hippocampus has a specifically high expression of FKBP51[16], it is highly sensitive to the effects of stress[17] and studies in MDD patients have demonstrated structural and cellular alterations in this region[18–20]. Moreover, the hippocampus is known to play a role in both emotional and spatial memory functions[21,22], two domains that are reflected in the symptomology of patients suffering from psychiatric disorders, such as MDD. The hippocampus has a diverse cell-type profile, but the excitatory glutamatergic pyramidal neurons make up the vast majority[23]. Notably, FKBP51 is particularly strongly expressed in excitatory neurons[24]. Previous work in human studies has identified *FKBP5* as a genetic risk factor for psychiatric disease[25,26], and studies in rodents have extensively described its role in stress vulnerability and resilience processes[27–32]. Interestingly, polymorphisms in the human *FKBP5* gene were found to interact with childhood trauma to increase the risk for developing psychiatric disorders[33–35]. The risk allele of the most studied *FKBP5* single nucleotide polymorphism (SNP) rs1360780 leads to a conformational change of the *FKBP5* DNA structure, and this causes the GRE in intron 2 to come in close contact with the transcriptional start site in the promotor region. Ultimately, this results in an enhanced glucocorticoid-induced FKBP51 induction and prolonged elevation of glucocorticoid concentrations after stress[36].

In mice, overexpression of FKBP51 in forebrain glutamatergic neurons led to differential effects of maternal separation on anxiety behaviour and hippocampal neurogenesis[28,37]. However, the mechanism underlying the interaction of FKBP51 variations with early-life adversity is not yet completely understood. Importantly, while stress-related psychiatric disorders are highly predominant in women[38–40], the consequences of ELS in females are severely understudied. To address this paucity, we here exposed both male and female mice with a deletion of FKBP51 in glutamatergic neurons to early life adversity and investigated the long-term behavioural, structural, functional and molecular consequences. Together, the data uncover a mechanism by which FKBP51 can contribute to a pro-resilient phenotype following ELS exposure.

## Results

### Fkbp5[Nex] genotype and ELS exposure have interactive effects on behaviour in female mice

In order to study the contribution of FKBP51 in glutamatergic forebrain neurons to the long-term consequences of ELS exposure, male and female mice lacking FKBP51 in glutamatergic neurons (*Fkbp5[Nex]*) and their *Fkbp5[lox/lox]* littermate WTs were exposed to an ELS paradigm from P2 to P9 or a control condition and underwent a behavioural experimental test battery in adulthood and subsequent structural MRI scanning (Fig. 1A). For females, ELS drastically reduced body weight (BW) at P9 at the end of the limited bedding and nesting (LBN) paradigm ($F_{(1, 36)} = 4.50$ $p < 0.001$), however no changes were found in body weight at the start of the experimental procedure, in adrenal weight or in baseline CORT levels at sacrifice (Supplementary Fig. 1A). For males, ELS also reduced body weight at P9 ($F_{(1, 40)} = 21.18$ $p < 0.001$) and in addition to this, a main effect of genotype was found for adrenal weight ($F_{(1, 29)} = 9.97$ $p < 0.01$) and CORT levels at baseline ($F_{(1, 29)} = 7.30$, $p < 0.05$.; Supplementary Fig. 1B).

The behavioural protocol included the open field (OF) and elevated plus maze (EPM) tests, assessing anxiety-like behaviour, and tests for (spatial) memory performance in a neutral environment - the spatial object recognition (SOR) and novel object recognition (NOR) - or in a stressful context - the Morris water maze (MWM). Data from the total distance travelled in the entire OF arena, for the complete 15-minute trial, revealed an increased locomotor behaviour for female *Fkbp5[Nex]* compared to *Fkbp5[lox/lox]* WTs, regardless of ELS exposure ($F_{(1,35)} = 7.38$, $p < 0.05$; Fig. 1B). Male mice were unaffected on general locomotor behaviour (Supplementary Fig. 2A). However, on the anxiety domain, behavioural changes were predominantly ELS-induced. Female mice that were exposed to ELS travelled a longer distance in the inner zone of the OF, for the first 10 min, independent of genotype ($F_{(1, 34)} = 7.50$, $p < 0.01$; Fig. 1B). Moreover, ELS-exposed females showed a strongly increased distance travelled in metres ($F_{(1, 35)} = 24.34$, $p < 0.001$) and time spent in seconds ($F_{(1,33)} = 24.84$, $p < 0.001$) in the open arms of the EPM (Fig. 1C). Notably, calculating the fold change (FC) of the ELS effect (mean OA distance = 1.39) vs. their respective control group (mean OA distance = 4.94) on EPM parameters revealed that the beneficial effect of ELS was significantly stronger in wild-type (WT) mice than in *Fkbp5[Nex]* mice (FC ELS effect EPM OA distance: $t_{(17)} = 2.30$, $p < 0.05$; Fig. 1D). These data suggest that ELS exposure leads to an anxiolytic phenotype in females, which is significantly dampened in *Fkbp5[Nex]* mice. In contrast to females, the phenotype induced by ELS and FKBP51 deletion was less pronounced in males. For males, a significant interaction effect was found on distance travelled in the inner zone of the OF ($F_{(1,38)} = 12.91$, $p < 0.05$: Supplementary Fig. 2A), but otherwise, no significant effects on anxiety-like behaviour were found (Supplementary Fig. 2B).

Interestingly, interaction effects between ELS exposure and *Fkbp5* genotype were also found for tests assessing memory performance in female mice. In a neutral environment, ELS exposure in females led to a worsened recognition memory, as defined by a lower discrimination of the novel object ($F_{(1, 25)} = 5.65$, $p < 0.05$; Fig. 1E), and an interaction effect between ELS and genotype was found for spatial memory function ($F_{(1, 31)} = 4.98$, $p < 0.05$; Fig. 1E). This was illustrated by a reduced discrimination between a familiar and novel location of the object (SOR), in which the effect of ELS on spatial memory was only present in *Fkbp5[Nex]* female mice. Furthermore, for the MWM, a spatial memory task performed under stressful circumstances, a clear interaction effect was observed as well. Remarkably, under the conditions used the female control mice did not learn the task as well as ELS-exposed females, but only in *Fkbp5[lox/lox]* mice (probe trial interaction effect: $F_{(1, 30)} = 4.63$, $p < 0.05$; Probe trial ELS effect: $F_{(1,30)} = 3.65$,

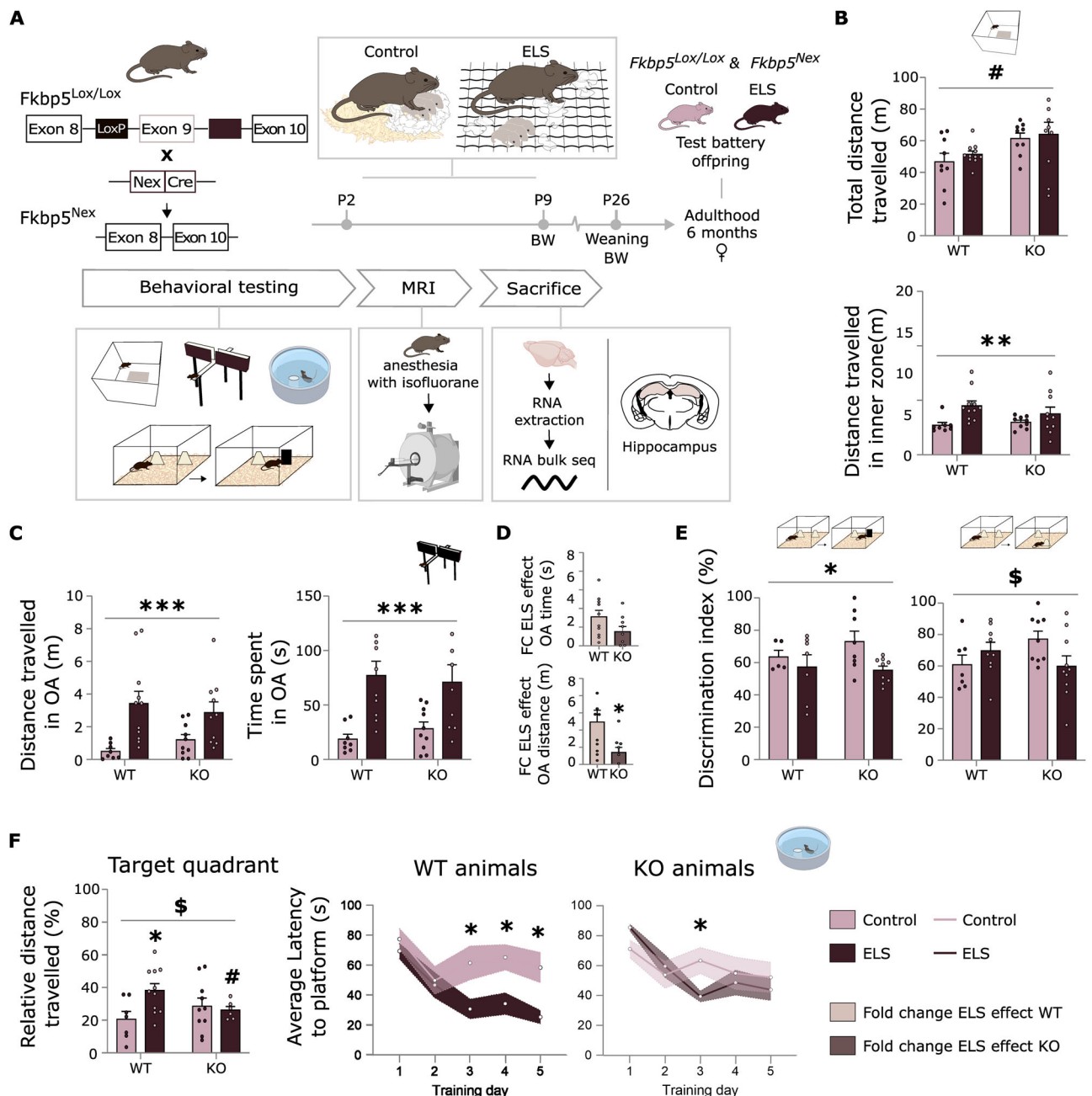

**Fig. 1 | FKBP51 in glutamatergic forebrain neurons and early life stress exposure have an interactive effect on behaviour in female mice. A** Offspring of *Fkbp5*^lox/lox^ WT mice and mice lacking FKBP51 in glutamatergic forebrain neurons (*Fkbp5*^Nex^) were exposed to limited bedding and nesting (LBN) early life stress (ELS) paradigm from postnatal day 2 (P2) to P9. At 6 months of age, mice were exposed to a behavioural protocol, including tests assessing anxiety-like behaviour and (spatial) memory performance in a neutral or stressful context. Panel (**B–F**) shows data of female mice only, data on male mice can be found in Supplementary Fig. 2. Genotype affected locomotor behaviour of female mice independent of ELS exposure (**B**). Data from the open field (OF) test (**B**) and elevated plus maze (EPM) test (**C**) revealed a strong beneficial effect of ELS exposure on anxiety-like behaviour. The fold change of the ELS effect (vs. the control condition) on EPM

parameters was significantly larger in WT mice than in KO mice (**D**). Further, memory performance in a neutral context (**E**) revealed a deteriorating effect of ELS, in interaction with FKBP5 genotype. Interestingly, for assessing memory function in a stressful context (**F**) with the Morris water maze (MWM), ELS had a beneficial effect on spatial memory performance in Fkbp5^lox/lox^ mice only. ELS: *n* = 10 *Fkbp5*^Nex^ and *n* = 11 *Fkbp5*^lox/lox^; control: *n* = 10 *Fkbp5*^Nex^ and *n* = 9 *Fkbp5*^lox/lox^. Error bars represent mean + S.E.M. Panels B,C,E,F: 2-way ANOVA. Panel (**D**): 2-sided *t* test. Panel (**F**): repeated measures ANOVA. *effect of ELS *p* < 0.05; **effect of ELS *p* < 0,01; *** effect of ELS *p* < 0.001; ^#^ effect of genotype *p* < 0.05; ^$^ interaction effect ELS x genotype *p* < 0.05. Images of MWM, mouse brain and MRT scanner created in BioRender. Schmidt, M. (2025) https://BioRender.com/p00d573.

*p* = 0.067; Fig. 1F). This interaction effect was also reflected in the average latencies to finding the platform location during the 5 training days. In *Fkbp5*^lox/lox^ mice, ELS-exposed females spent significantly less time to finding the platform on the last three training days compared to control mice (post-hoc day 3: *p* < 0.05; day 4: *p* < 0.05; day 5: *p* < 0.05; Fig. 1F). This strongly improved memory function following

ELS was not observed in *Fkbp5*^Nex^ mice (difference only on training day 3: *p* < 0.05). Male mice did not show differences on memory performance following ELS exposure, nor were any interactive effects observed (Fig S2C). To confirm the validity of the MWM protocol in females, a control cohort was tested with an adjusted inter-trial interval, rendering the cognitive task slightly easier to acquire. Under

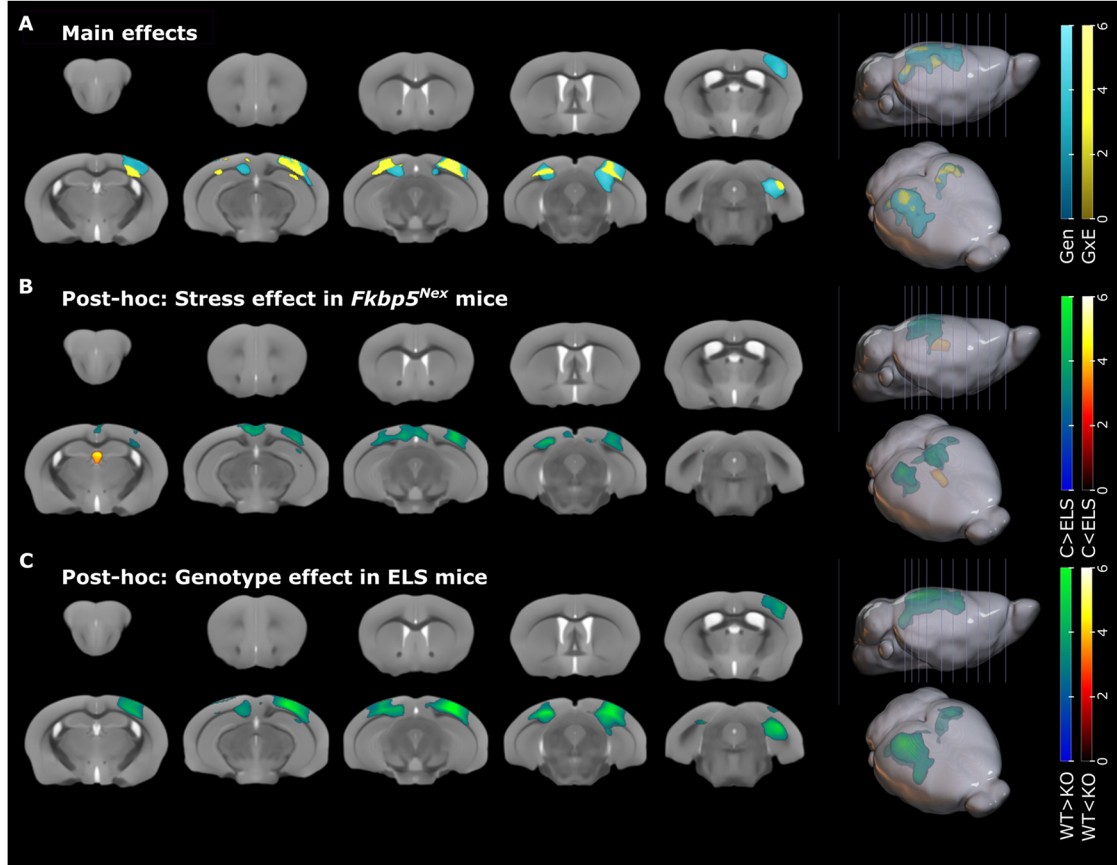

**Fig. 2 | ELS and *Fkbp5* genotype lead to separate and interactive changes in brain volume in female mice. A** Deformation-based morphometry analyses of the female brains revealed main effects of genotype (Gen, T-contrast WT > KO) in grey matter (GM) volumes of the cortex (right somatosensory cortex and bilateral visual cortex) and in the ventral subiculum. Overall white matter (WM) deformations were found for genotype in the bilateral dorsal hippocampal commissure. In addition, interactive effects of *Fkbp5* genotype and early life stress (ELS) exposure (GxE, T-contrast, pronounced negative effect of stress in KO mice compared to WT), were found in various cortical regions (right somatosensory cortex, bilateral visual cortex and bilateral retrosplenial cortex). **B** Post-hoc *t* tests revealed reductions in brain volume following ELS, in *Fkbp5^Nex^* mice (KO) only, within bilateral cortical areas and increased volumes in the third ventricle. **C** Furthermore, within mice that underwent ELS, *Fkbp5^Nex^* mice had smaller GM volumes in cortical areas and reduced volumes in WM structures of the dorsal hippocampal commissure. Scales represent Z-scores. WT = wild-type, KO = Fkbp5^Nex^, C = control, ELS = early life stress, Gen = genotype, GxE = gene by environment interaction. ELS: *Fkbp5^lox/lox^* $n = 7$, *Fkbp5^Nex^* $n = 10$;control: *Fkbp5^lox/lox^* $n = 11$ and *Fkbp5^Nex^* $n = 8$.

these conditions, also non-stressed WT females were able to significantly reduce the latency to find the hidden platform over the course of the training days (Supplementary Fig. 2D).

## Fkbp5^Nex^ genotype interacts with ELS exposure to affect brain volume in female mice

We next probed potential structural and functional differences in these mice. To determine whether the observed changes in behaviour are accompanied by volumetric brain differences, mice of both sexes underwent an MRI structural scan subsequent to behavioural testing. In females, a two-way ANOVA deformation-based morphometry (DBM) analysis was performed and revealed main effects of genotype in GM structures in the right somatosensory cortex, the bilateral visual cortex and in the ventral part of the subiculum (*T* test WT > KO; right cluster $p_{FWE, cluster} < 0.001$ and left cluster $p_{FWE, cluster} = 0.007$; Fig. 2A). Furthermore, white matter (WM) deformations were found in the bilateral dorsal hippocampal commissure (right cluster $p_{FWE, cluster} < 0.001$ and left cluster $p_{FWE, cluster} = 0.007$; Fig. 2A). In addition to this, a main interaction effect was found within these regions, with exception of the somatosensory cortex (*T* test; right hemisphere: $p_{FWE, cluster} = 0.007$; left hemisphere: $p_{uncorrected cluster} = 0.024$; Fig. 2A). Post-hoc *t* tests showed that ELS exposure, within the *Fkbp5^Nex^*, led to reduction in grey matter (GM) volume in the right somatosensory and

visual cortex ($p_{FWE, cluster} = 0.007$). This cluster extends into the right caudal hippocampus. A further volumetric reduction was found for the left visual cortex and retrosplenial area ($p_{FWE, cluster} = 0.005$; Fig. 2B). ELS, however, showed a tendency to increase the volume of the third ventricle ($p_{uncorrected, cluster} = 0.039$). In addition, within female mice that were exposed to ELS, *Fkbp5^Nex^* mice compared to *Fkbp5^lox/lox^* mice, had reduced volumes in the right somatosensory cortex, bilateral visual cortex, left retrosplenial area, the ventral subiculum and WM structures of the dorsal hippocampal commissure (right cluster $p_{FWE, cluster} < 0.001$, left cluster $p_{FWE, cluster} = 0.007$; Fig. 2C).

Two-way ANOVA DBM analysis in male mice revealed the main effects of genotype in the bilateral subiculum ($p_{uncorrected cluster} < 0.005$). The main effect of stress was visible in the periaqueductal grey (PAG) ($p_{FWE, cluster} = 0.007$), while an interaction effect genotype x stress was observed in the right somatosensory cortex ($p_{FWE, cluster} = 0.041$) and in the septal nucleus extending to the anterior forceps of the corpus callosum ($p_{FWE, cluster} = 0.041$). Post-hoc *t* tests showed that ELS exposure, within the *Fkbp5^Nex^*, led to reduced volume in the retrosplenial cortex ($p_{uncorrected cluster} = 0.025$) and in the right hippocampus ($p_{uncorrected cluster} = 0.046$), whereas *Fkbp5^lox/lox^* mice showed larger volume in the right somatosensory cortex ($p_{FWE, cluster} < 0.001$) in stressed mice as compared with control animals. The left somatosensory cortex did not reach cluster significance

($p_{uncorrected\ cluster}$ = 0.068). In *Fkbp5^{lox/lox}* mice, the PAG showed reduced volume for ELS mice ($p_{uncorrected\ cluster}$ = 0.005, with $p_{FWE,\ cluster}$ = 0.064), as did the right BNST and right somatosensory cortex ($p_{uncorrected\ cluster}$ < 0.05). The genotype effect in stressed mice was visible as large somatosensory cortical volumes in *Fkbp5^{lox/lox}* mice ($p_{FWE,\ cluster}$ = 0.001 (right) and 0.023 (left)), whereas as cluster extending from the BNST and septal nucleus to the piriform area showed larger volume in *Fkbp5^{Nex}* mice ($p_{FWE,\ cluster}$ = 0.001). Finally, unstressed control mice showed larger volume in the right anterior forceps and the dorsal part of the corpus callosum ($p_{uncorrected\ cluster}$ = 0.01; Supplementary Fig. 3).

### Fkbp5^{Nex} genotype and ELS exposure have interactive effects on hippocampal dendritic morphology and synaptic function in female mice

To investigate the underlying mechanisms of the interactive effects between ELS exposure and *Fkbp5* genotype, that we observed particularly on memory performance, we investigated the neuronal structure of pyramidal neurons in the CA1 area of the hippocampus, by staining brains of a separate cohort of female *Fkbp5^{Nex}* and *Fkbp5^{lox/lox}* mice that underwent ELS exposure or a control condition (Fig. 3A) and defined both the spine densities. Interestingly, a very similar interactive effect as was observed for cognitive behaviour in a stressful context was also reflected in the CA1 pyramidal neuron structure (apical spines: interaction effect $F_{(1, 48)}$ = 8.84, $p < 0.01$, main effect ELS $F_{(1, 48)}$ = 30.99 $p < 0.001$; basal spines: interaction effect $F_{(1, 48)}$ = 5.09, $p < 0.05$, main effect ELS $F_{(1, 48)}$ = 23.98, $p < 0.001$; Fig. 3A). ELS exposure increased the number of both apical ($p < 0.001$) and basal spines ($p < 0.001$) compared to the control condition in *Fkbp5^{lox/lox}* mice, but did not change the number of spines in *Fkbp5^{Nex}* mice. An interaction effect and main effect for ELS exposure was also found for complexity in dendritic branching in the CA1 region of the hippocampus, measured as the number of dendritic intersections (sum of dendritic intersections: interaction effect $F_{(1,121)}$ = 15.73, $p = 0.001$, main effect of ELS $F_{(1, 121)}$ = 69.43, $p < 0.001$; Fig. 3B). ELS exposure leads to increases in dendritic complexity in both *Fkbp5^{lox/lox}* ($p < 0.001$) and *Fkbp5^{Nex}* mice ($p = 0.01$). For males, no significant main or interaction effects for CA1 pyramidal neuron structure were observed (Supplementary Fig. 4A, B).

Next, to test for potential changes in synaptic function, we performed field potential recordings at dorsal CA3-CA1 synapses in acute brain slices and examined their ability to express high-frequency stimulation (HFS)-induced long-term potentiation (LTP) under different conditions. For females, we first investigated the interactive effects of *Fkbp5* genotype and an ex vivo stress exposure (Fig. 3C), in the form of a corticosterone application to the brain sections vs. a vehicle condition. Again, we observed an interactive effect between stress exposure and genotype (interaction effect genotype x CORT: $F_{(1, 32)}$ = 16.84, $p < 0.001$). CORT application led to a reduction in LTP in *Fkbp5^{lox/lox}* mice ($p < 0.001$), which was absent in *Fkbp5^{Nex}* mice. Following up on this, we studied the effects of ELS exposure in females vs. control mice on LTP. Strikingly, the interactive effect between *Fkbp5* genotype and ELS exposure that was already observed on the behavioural and structural neuronal level, was again confirmed on the electrophysiological level. We found a main effect of ELS exposure on reducing LTP ($F_{(1, 36)}$ = 7.63 $p < 0.01$; Fig. 3D) and an interaction effect between ELS exposure and genotype ($F_{(1, 36)}$ = 6.49 $p < 0.05$). Post-hoc tests revealed that ELS decreased LTP in *Fkbp5^{lox/lox}* mice ($p < 0.01$), but not in *Fkbp5^{Nex}* mice (Fig. 3D). In males, we also tested for interactive effects of *Fkbp5* genotype and an ex vivo corticosterone exposure (Supplementary Fig. 4C). Interestingly, here we also observed an interactive effect between stress exposure and genotype (interaction effect genotype x CORT: $F_{(1, 24)}$ = 4.741, $p < 0.05$). CORT application led to a reduction in LTP in Fkbp5^{lox/lox} mice ($p < 0.001$), which was absent in Fkbp5^{Nex} mice. In summary, we observed robust changes upon ELS exposure in WT *Fkbp5^{lox/lox}* mice, with beneficial outcomes on the behavioural level, that were not present in mice lacking *Fkbp5* in glutamatergic forebrain neurons. This interactive nature between *Fkbp5* genotype and (beneficial) ELS effects implies that FKBP51 in glutamatergic forebrain neurons may play a role in the changes that are happening in the brain during stress exposure in early developmental stages.

### RNA bulk sequencing reveals TCF4 as a potential target for molecular pathways underlying the interactive effects between Fkbp5^{Nex} genotype and ELS exposure in females

Our analyses revealed beneficial effects of ELS exposure on behaviour, specifically in female WT *Fkbp5^{lox/lox}* mice, that were absent in mice lacking FKBP51 in the glutamatergic neurons of the forebrain. In addition, the interaction between the *Fkbp5* genotype and ELS exposure was further emphasised in CA1 neuronal structure and electrophysiological properties. These findings suggest an important mediating role for glutamatergic forebrain FKBP51 in the alterations that exposure to stress during early life has on the brain. However, the exact underlying molecular mechanisms still remained unclear. To further unravel molecular processes that may underlly these observed changes, we performed bulk mRNA sequencing in the hippocampus of female *Fkbp5^{Nex}* and *Fkbp5^{lox/lox}* mice. The data revealed a differential expression pattern that was associated with the effect of genotype (13 DEGs upregulated in *Fkbp5^{Nex}* and 33 DEGs downregulated in *Fkbp5^{Nex}*; Fig. 4A). To better explore the contribution of networks of genes that might impact behaviour, brain structure and function, we performed a weighted gene co-expression network analysis (WGCNA)[41] and identified a total of 18 co-expressed gene modules (Fig. 4B). Five of these modules were nominally associated with the genotype effect, 9 with the ELS exposure and the remaining 5 with the interaction of *Fkbp5* genotype and ELS exposure (Supplementary Table 1). Interestingly, there was one specific module (darkorange, 176 genes), that was not only associated with ELS exposure, but was additionally associated with the interaction of *Fkbp5* genotype and ELS.

The previous interactive findings between *Fkbp5* genotype and ELS exposure on behaviour, brain structure and function were mainly driven by the ELS effect within WT *Fkbp5^{lox/lox}* mice and the absence of this effect in *Fkbp5^{Nex}* mice. Therefore, the dark orange co-expressed gene module was particularly interesting and was selected for further in-depth analyses. Especially the identified hub genes in this module had a similar direction of effect, with increased expression levels upon ELS exposure in *Fkbp5^{lox/ox}* mice, but no increases in expression within *Fkbp5^{Nex}* mice (Supplementary Fig. 5). Subsequently, we performed a transcription factor enrichment analyses in the dark orange module (Fig. 4C) to find transcription factors that may be the driving force behind this important module of genes. This revealed a total of 10 transcription factors that at least regulated 8 of the 176 genes in the module (Supplementary Fig. 6). To decide which of these transcription factors could be the most important driver of the module in the light of our previous findings, we compared our datasets with other relevant human GWAS datasets from PTSD patients and individuals (both sexes and females only) that had suffered from childhood trauma (Fig. 4D). Furthermore, we also overlaid the selected genes with our list of hub genes of the dark orange module, to identify genes that are highly interconnecting with other genes in the module and may therefore have a strong driving force. From the genes that were regulated by any of the enriched transcription factors, 8 were also found significantly affected in one of the GWAS datasets or the hub genes dataset (Fig. 4D and Supplementary Fig. 6). The transcription factor that regulated most of the genes that had an overlap with the selected datasets (5 overlap hits; *Foxp2*, *Slc17a6*, *Tcf7l2*, *Zic1* and *Zic4*), was the transcription factor 4 (TCF4) (Fig. 4D). Moreover, this transcription factor regulated the only gene that was associated with childhood trauma in females (*Slc17a6*) and it regulated three hub genes in the dark orange module

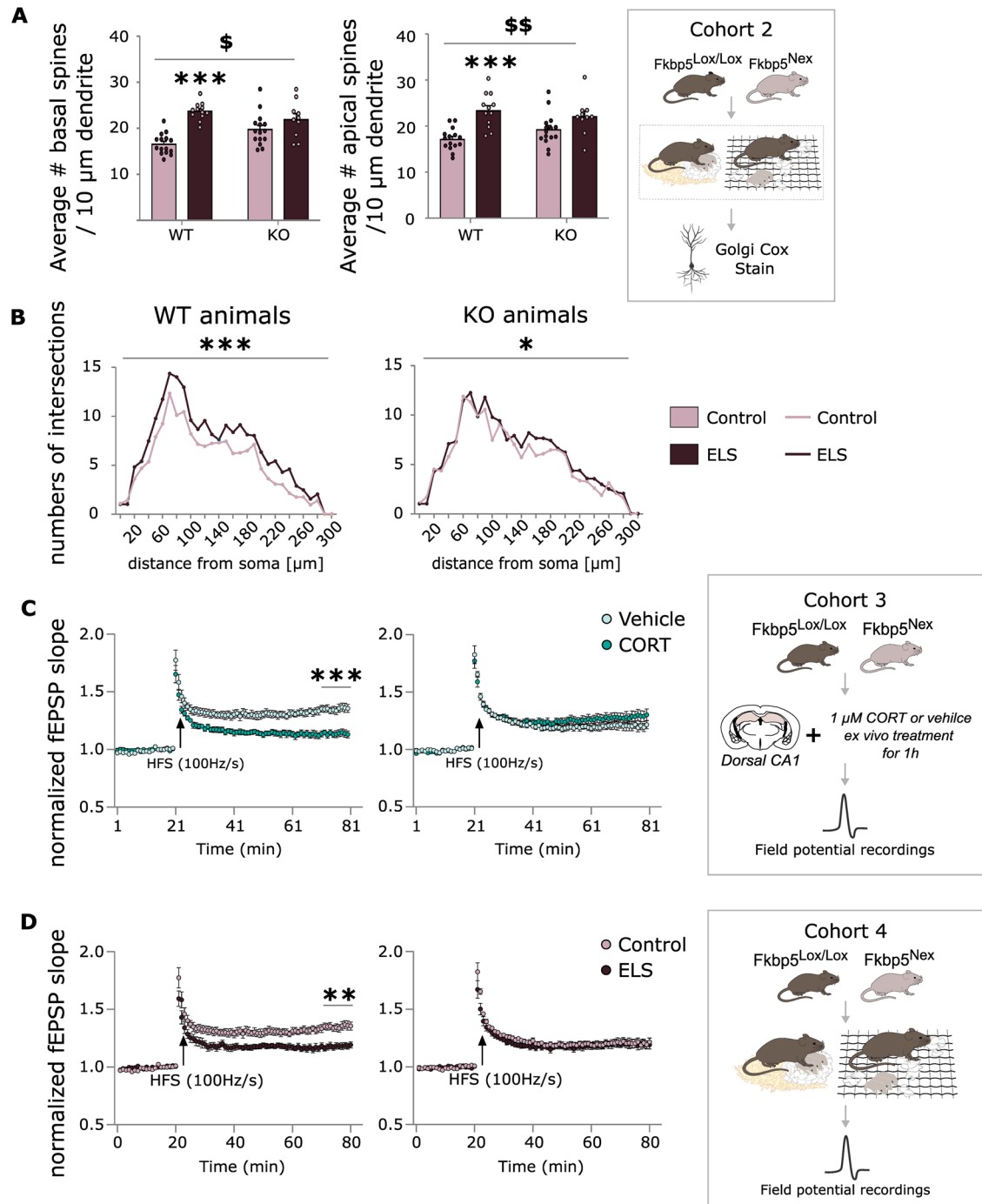

**Fig. 3 | FKBP51 in glutamatergic forebrain neurons and early life stress exposure have an interactive effect on neuronal structure and function in female mice.** A second cohort (**A**) of female *Fkbp5^Nex* and *Fkbp5^lox/lox* offspring was exposed to limited bedding and nesting (LBN) early life stress (ELS) paradigm, and a Golgi Cox staining was performed at the hippocampus of the 8-month old female mice. A very similar interaction effect, as was observed for behaviour, was also found for apical spine densities in pyramidal neurons of the CA1 region (ELS: *Fkbp5^Nex* $n = 15$ neurons vs. *Fkbp5^lox/lox* $n = 12$ neurons; Control: *Fkbp5^Nex* $n = 15$ neurons vs. *Fkbp5^lox/lox* $n = 15$ neurons). Moreover, an interaction effect and main effect of ELS was found for dendritic complexity (**B**), where dendritic complexity of CA1 pyramidal neurons was increased following ELS exposure, particularly in *Fkbp5^lox/lox* mice (ELS: *Fkbp5^Nex* 30 neurons vs. *Fkbp5^lox/lox* 40 neurons; Control: *Fkbp5^Nex* 21

neurons vs. *Fkbp5^lox/lox* 33 neurons). In addition, data from electrophysiological LTP measurements, again demonstrated interactive effects of the *Fkbp5* genotype with an ex vivo stress exposure, in the form of a corticosterone application (**C**; CORT: *Fkbp5^Nex*: $n = 4$ mice, $n = 11$ brain slices; *Fkbp5^lox/lox*: $n = 5$ mice, $n = 11$ brain slices, vehicle: *Fkbp5^Nex*: $n = 5$ mice, $n = 9$ brain slices; *Fkbp5^lox/lox*: $n = 6$ mice, $n = 9$ brain slices), and with ELS exposure (**D**; ELS: *Fkbp5^Nex*: $n = 4$ mice, **n** = 11 brain slices; *Fkbp5^lox/lox*: $n = 5$ mice, $n = 11$ brain slices; control: *Fkbp5^Nex*: $n = 5$ mice, $n = 9$ brain slices; *Fkbp5^lox/lox*: $n = 6$ mice, $n = 9$ brain slices). Error bars represent mean + S.E.M. Panel (**A**): 2-way ANOVA. Panel (**B**, **C**): repeated measures ANOVA. **effect of ELS $p < 0,01$; *** effect of ELS or CORT $p < 0.001$; $ interaction effect ELS x genotype $p < 0.05$; $$ interaction effect ELS x genotype $p < 0.01$. Image of neuron created in BioRender. Schmidt, M. (2025) https://BioRender.com/o64z476.

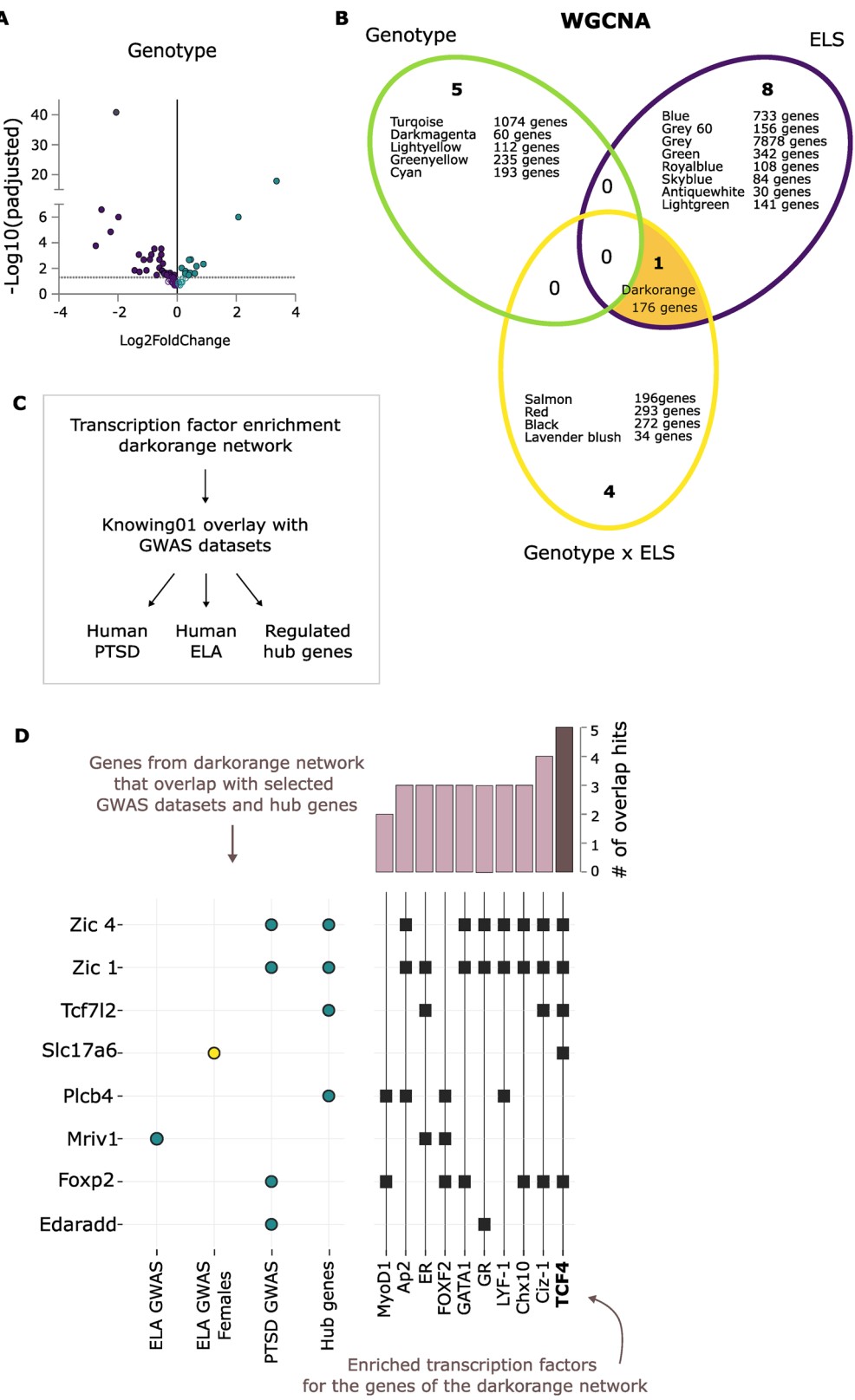

(*Tcf7l2* and *Zic1* and *Zic4*). A direct regulation of hippocampal TCF4 itself was not observed, and this absence of regulation was confirmed by qPCR (Supplementary Fig. 7A). We also tested for an involvement of the important stress regulators MR and GR in the hippocampus, but also found no significant expression differences as a consequence of ELS or FKBP51 deletion (Supplementary Fig. 7B). Based on these findings, TCF4 could be a potential interesting factor, responsible for

underlying mechanisms of the interactive effects that we observed between ELS exposure and *Fkbp5* genotype.

To test for the hippocampal expression of TCF4, identified TCF4 target genes, dark orange module hub genes, and the highly stress-relevant receptors MR and GR in male *Fkbp5*$^{Nex}$ or *Fkbp5*$^{lox/lox}$ mice with or without a history of ELS, qPCR analyses were performed (Supplementary Fig. 8). While we observed an interesting pattern of gene

**Fig. 4 | RNA bulk sequencing reveals transcription factor 4 as a potential regulator of early life stress-induced effects on the brain that interact with FKBP51 in glutamatergic forebrain neurons.** Bulk mRNA sequencing was performed on the hippocampus of female mice of the first cohort. **A** A clear differential expression profile was found for the effects of genotype. Furthermore, a weighted gene co-expression analysis (WGCNA) revealed 18 co-expressed gene modules that were associated with effects of genotype, ELS exposure or their interaction (**B**). One of these modules was associated not only with ELS exposure but was also associated with the interaction of ELS and *Fkbp5* genotype. **C** Subsequently, a transcription factor enrichment analysis was performed for the dark orange module, which

resulted in 10 enriched transcription factors. Using the software Knowing01, all genes that are regulated by the enriched transcription factors were overlaid with datasets from human psychiatric GWAS studies and the hub genes of the dark orange module (**D**). The right panel (**D**) shows which genes are regulated by specific enriched transcription factors and their resulting (indirect) overlap with the datasets. This revealed that the transcription factor 4 (TCF4) regulates the largest number of genes that had an overlap with any of the datasets. Moreover, it is the only enriched transcription factor that regulates a gene that was associated with early life adversity in females (Slc17a6; yellow dot).

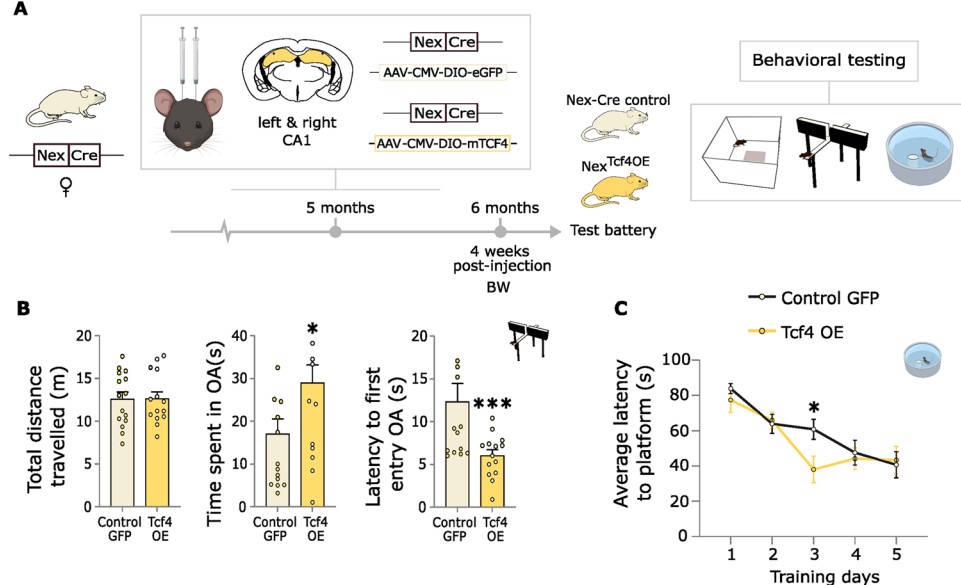

**Fig. 5 | Tcf4 overexpression leads to similar beneficial effects on behaviour as ELS exposure in *Fkbp5<sup>lox/lox</sup>* mice.** To investigate whether TCF4 in glutamatergic neurons of the hippocampus is indeed the underlying molecular target regulating the beneficial effects of ELS exposure on behaviour, we overexpressed *Tcf4* in these neurons by injecting an AVV Cre-dependent *Tcf4* overexpression (OE) virus in Nex-Cre female mice of 6 months of age and compared them to female mice that were injected with an AVV GFP control virus (**A**). Interestingly, TCF4 OE indeed leads to (**B**) an anxiolytic phenotype on the elevated plus maze (EPM) test, like was observed

upon ELS exposure in *Fkbp5<sup>lox/lox</sup>* mice. Furthermore, data from the MWM showed that TCF4 OE mice had an improved memory of the platform location on the 3rd training day, which was a less strong but similar effect to as was observed for ELS-exposed female *Fkbp5<sup>lox/lox</sup>* mice. n = 15 per group. Error bars represent mean + S.E.M. Panel B: Wilcoxon signed-rank test. Panel (**C**): repeated measures ANOVA. *p < 0.05; **p < 0,01; ***p < 0.001. Images of mouse head and MWM created in BioRender. Schmidt, M. (2025) https://BioRender.com/o64z476.

expression across these target genes, with an altered expression following ELS in *Fkbp5<sup>Nex</sup>*, but not *Fkbp5<sup>lox/lox</sup>*, mice, these effects did not reach statistical significance in males.

## Viral overexpression of Tcf4 in glutamatergic neurons of the hippocampus leads to changes on behaviour and TCF4-driven gene expression

From the bulk mRNA sequencing data and follow-up analyses, TCF4 was identified as a potentially important regulator of the ELS-induced effects that were observed in *Fkbp5<sup>lox/lox</sup>* females but were absent in *Fkbp5<sup>Nex</sup>* females. However, a direct link between the ELS induced-phenotype and over-stimulation of the dark orange module by TCF4, remained speculative. Based on the strongest interactive phenotype in stressed cognition in the previous cohort, we overexpressed Tcf4 in the CA1, specifically in the glutamatergic neurons by bilateral injections with a Cre-dependent *Tcf4* OE AVV virus vs. a GFP expressing control AVV virus in 5 months-old female mice (Fig. 5A and Supplementary Fig. 9). Interestingly, in line with what we found for *Fkbp5<sup>lox/lox</sup>* mice that were exposed to ELS, we found that mice that had an OE of *Tcf4* in the glutamatergic neurons of the hippocampus showed reduced anxiety-like behaviour (Fig. 5B). This was reflected by an increased time spent in seconds in the open arms on the EPM (*W* = 64, *p* < 0.05) and a reduced latency towards the first entry into the open

arm in seconds (*W* = 149.5, *p* < 0.001), whereas locomotor behaviour was unaffected. Moreover, Tcf4 OE also had a similar effect on spatial memory performance under stressful conditions as ELS exposure (Fig. 5C). We found that on the 3rd training day of the MWM, Tcf4OE mice had significantly shorter latencies to finding the platform location than mice injected with the control virus (repeated measures ANOVA; post hoc (t): *p* < 0.05), indicating an improved learning of the spatial location. However, control animals did catch up to Tcf4 OE mice on the 4th training day. These data indicate that TCF4 in the glutamatergic neurons of the hippocampus, at least in part, contributes to the beneficial effects that ELS exposure has on behaviour.

In a separate cohort of female and male mice with or without CA1 overexpression of TCF4, we again performed qPCR analyses for TCF4 and the corticosterone receptors MR and GR. Again, neither of the glucocorticoid receptors was changed in expression due to TCF4 overexpression in females or males (Supplementary Fig. 10).

## Discussion

Psychiatric disorders often arise as a combination of environmental and genetic factors, and early life adversity has frequently been described as a risk factor for developing psychiatric disease[2,42]. Nevertheless, ELS can also lead to adaptive changes that prepare an individual to cope with future life events[11,12]. The *FKBP5* gene is a

psychiatric risk factor that is known to interact with ELS exposure[35,36]. However, the exact underlying mechanisms behind this interactive effect are still poorly understood. Moreover, FKBP51 functionality is cell-type specific and largely dependent on sex[43]. Unfortunately, up to date, there is still a scarcity of information on the effects of ELS and FKBP51 functionality in the female sex. This study demonstrated that FKBP51 in glutamatergic forebrain neurons mediates (beneficial) effects of an ELS exposure on emotional regulation, cognitive functioning and brain volume, particularly in females, and that this was associated with similar interactive effects on neuronal structure and function. Furthermore, we propose TCF4 as an underlying regulator of the FKBP51-mediated effects of ELS exposure on the brain and behaviour.

In this study, we provide evidence that ELS results in anxiolytic behaviour and improves spatial memory performance in a stressful context, and this effect is dependent on the presence of FKBP51 in glutamatergic forebrain neurons. The hyperactivity in a novel environment observed specifically in female $Fkbp5^{Nex}$ mice unlikely underlies this phenotype, as it was not mediated by ELS exposure. Interestingly, the FKBP51-dependent beneficial effects of ELS are predominantly present in female mice, while at a structural and neuronal functional level both sexes are affected, albeit distinctively. A body of research is available on the effects of ELS exposure on brain and behaviour[44–48] and the majority of these findings show that ELS can lead to negative outcomes on brain structure, brain function and behaviour. However, the effects of ELS are highly dependent on a number of factors, amongst others the age of the animal, the type, severity or duration of the early life stressor or the context in which the test takes place[7,48,49]. A number of studies in rodents, primates and humans have shown that exposure to ELS can also result in beneficial alterations in brain function, neuroendocrine responses and behaviour[7–10]. Moreover, an extremely important factor to consider is the sex of the animal[50–52]. In the past decades, it has become increasingly clear that sex can have a tremendous effect on stress resilience and vulnerability. Nevertheless, there is still a large gap in research in females and studies investigating the effect of ELS exposure are no exception. In a recent meta-analysis on early life adversity, Joëls and colleagues were unable to perform a quantitative analysis on the female data, due to a scarce availability of female studies[53]. It is, therefore, not strange that ELS in females might result in differential outcomes as traditionally described in males. FKBP51 functionality has also been shown to be heavily dependent on sex[43]. In line with our results, previous work has already demonstrated a sex-dependent interaction of FKBP51 and ELS on emotional behaviour. A study by Criado-Marrero and colleagues found that overexpression of FKBP51 amplified the anxiogenic effects of maternal separation stress, and this effect was more pronounced in female mice[37]. Unlike the study by Criado-Marrero and colleagues, we found beneficial effects on anxiety and cognition in a stressful context following ELS exposure, but we also found the effects to be predominantly present in female mice. The differences in outcomes on behaviour between our study and the study by Criado-Marrero and colleagues might be explained by the use of a different ELS paradigm. On the other hand, Maniam and Morris reported that female rat offspring exposed to non-handling were more resilient and displayed fewer adverse behavioural responses than the male rats[54]. Interestingly, we found opposing effects of ELS exposure on cognitive behaviour, depending on the context the test was performed. Unlike for spatial memory functioning in a stressful context, we found that in a neutral environment, ELS lead to a worsened memory function. Notably, opposing to findings from the stressful memory task, effects were exacerbated in $Fkbp5^{Nex}$ mice. Such dependency on the environmental context has been described previously in relation to cognitive behaviour[55,56]. In fact, these findings would be in line with the stress inoculation hypothesis that proposes that moderate ELS prepares for future matching life events. Illustrating

this, in this study, the ELS exposure resulted in detrimental effects in an "unmatched environment" (memory task in neutral context), but in beneficial effects in a "matched environment" (stressful memory task)[11] compared to the unstressed control group. Nonetheless, more studies on sex differences of ELS effects, ideally with a combined-sex design with head-to-head comparisons, are needed across different species to strengthen the translational potential of our findings and inform the development of targeted interventions for stress-related disorders.

In addition to the changes in behaviour, we also observed interactive effects of ELS and FKBP51 in glutamatergic forebrain neurons on brain volume in male and female mice. In females, ELS was found to result in volumetric reductions in several GM and WM structures, amongst others, in different cortical regions, the ventral subiculum and the dorsal hippocampal commissure and these effects of ELS were exclusively present in $Fkbp5^{Nex}$ mice. Spatial memory function is strongly linked to activity in the dorsal hippocampus[22]. However, the MRI results indicated that our behavioural effects in female mice could not directly be linked to GM volume changes in this brain region. Nevertheless, volumetric alterations in the fibre tracts of the dorsal hippocampal commissure in stressed $Fkbp5^{Nex}$ mice can also be linked to memory function[57]. The dorsal hippocampal commissure is a WM structure that is responsible for interhemispheric connections between the temporal regions[57]. It, therefore, plays an important role in the communication between the hippocampus and other temporal lobe regions, such as the amygdala, a brain region that is heavily implicated in fear and anxiety behaviour. It was previously demonstrated that the dorsal hippocampal commissure was associated with recognition memory[57]. Therefore, volumetric reductions in this WM tract would match with the observed impaired memory performance on the novel object recognition task, an effect that was specifically present in ELS-exposed $Fkbp5^{Nex}$ female mice. Alternatively, one could speculate that the adaptive changes by ELS, resulting in enhanced spatial memory performance in a stressful context, are not occurring in $Fkbp5^{Nex}$ mice, and this is reflected by the observed reduced GM and WM volumes. Another interesting finding was the reductions in the ventral subiculum in ELS-exposed $Fkbp5^{Nex}$ female mice. The subiculum is most commonly known as an integrator for the output of hippocampal information to other brain regions, however, it has a segregated functionality along the dorsoventral axis[58]. Where the dorsal part of the subiculum is thought to be involved in the processing of spatial memory information, the ventral subiculum is implicated in HPA-axis feedback. More specifically, via glutamatergic ventral subiculum output neurons, the hippocampus dampens the stress-induced glucocorticoid release, by connecting to neurons in the paraventricular nucleus of the hypothalamus[59]. It is, therefore, unsurprising that interactions between ELS exposure and FKBP51 in glutamatergic neurons, whose primary function is to regulate GR sensitivity, are associated with changes in this region. Similarly, in males, structural alterations related to ELS, FKBP51 in glutamatergic neurons, or an interaction of both factors were observed, which partially overlapped with the effects detected in females, specifically for cortical regions. However, in males also, distinct regions were structurally affected, including the piriform cortex and the periaqueductal grey, pointing towards male-specific effects that may also underlie the differences in their behavioural phenotype.

Complementing the interactive findings of ELS and FKBP51 in glutamatergic forebrain neurons on memory performance, we found highly similar interactive patterns on dorsal CA1 pyramidal neuronal structure and function. ELS improved spatial memory performance in a stressed context in WT female mice and, strikingly, ELS also exclusively increased the spine density in CA1 pyramidal neurons in female, but not male, $Fkbp5^{lox/lox}$ mice. Furthermore, we found that an ex vivo glucocorticoid administration reduces LTP in the dorsal CA1 region in WT mice, but not in $Fkbp5^{Nex}$ mice of both sexes. The same was true for ELS exposure, which was only assessed in females. Even though at this

point, the observed female-specific improved stress-related cognition, increased spine density and decreased synaptic plasticity remain elusive, the direct dependence of these stress-induced alterations on glutamatergic FKBP51 function is highly apparent. Our data clearly indicate that independent of the directionality, early-life adversity effects in female mice are dependent on FKBP51 in glutamatergic neurons. The data further support previous findings that reduced synaptic transmission can be associated with enhanced spatial memory performance[60] and that ELS effects on cognition and LTP are highly dependent on the test context[7].

A potential mechanism underlying the effects of FKBP51 with regard to ELS is its functional interaction with the glucocorticoid receptors GR and MR, but potentially also other steroid receptors, including those for sex steroids. While we did not assess the influence of sex hormone signalling in the current study, the important influence of these hormones cannot be neglected[61] and should be included in future studies. The expression level of MR and GR was found to be unaltered in the current study, but this does not exclude a more subtle functional contribution of these steroid receptors.

Based on the findings from the RNA sequencing, we identified one module of genes that could be the driving force behind of the FKBP51-mediated effects of ELS on the behaviour of female mice. Interestingly, we found TCF4 to be an important enriched transcription factor of this module of genes. This finding is independent of TCF4 expression itself, which we confirmed in independent cohorts of males and females not to be regulated by ELS or FKBP51 expression under basal conditions. However, this does not exclude a difference in the dynamic regulation of this transcription factor at specific time points on the protein level or via post-translational modifications. Indeed, a follow-up study showed that enhanced TCF4 activity in glutamatergic neurons of the hippocampus, on its own, is sufficient to induce highly similar beneficial effects on anxiety and spatial memory in females in a stressful context as was observed with ELS exposure.

Thus, we propose TCF4 as an underlying regulator of the observed beneficial FKBP51-mediated ELS effects. TCF4 belongs to the helix-loop-helix protein family that can bind DNA as homo- or heterodimers at the E-box sites and thereby regulate transcription of a number of target genes[62]. It has, amongst others, been implicated in neurogenesis[63], been shown to affect neuronal morphology[64] and is involved in memory and learning processes and associated neuronal activity[65]. More specifically, mice with a knockdown of TCF4 were presented with spatial memory deficits on the MWM and this was accompanied by improved LTP in the CA1 region of the hippocampus. Furthermore, TCF4 has been associated with oligodendrocyte functioning and myelination processes[66], which could reflect the observed changes in WM structures. Intriguingly, a recent study observed a negative correlation of TCF4 with FKBP5 expression in muscle tissue of patients with type-2 diabetes[67]. In humans, TCF4 has already frequently been implicated in a number of psychiatric and neurological disorders such as schizophrenia, bipolar disorder, MDD, PTSD and autism[62,68]. Interestingly, childhood maltreatment is a major risk factor for MDD and PTSD[69,70]. Thus, we here provide evidence that TCF4-mediated transcriptional regulation, specifically in females, might drive a pro-resilient phenotype.

Taken together, this study showed that, particularly in females, ELS has adaptive effects on behaviour by inducing structural and functional changes in the hippocampus. These underlying alterations in neuronal morphology and electrophysiological properties of CA1 pyramidal neurons and GM and WM changes in cortical and subcortical regions are dependent on the presence of FKBP51 in glutamatergic neurons. In part, these FKBP51-dependent changes could be regulated via an augmented transcriptional drive of a network of genes by the TCF4 transcription factor, but a causal relationship between TCF4 and FKBP51 still remains to be established. This study provides insights into how ELS can affect behaviour in an adaptive manner and

proposes FKBP51 and TCF4 as highly interesting targets for further research towards mechanisms of ELS resilience. Small molecule inhibitors for both proteins have already been described[71], but whether these can be targeted in a cell-type-specific fashion still remains a challenge.

## Method

### Animals and housing conditions

All animals were bred at the in-house breeding facility of the Max Plank Institute of Psychiatry in Martinsried, Munich, DE. Unless specifically stated otherwise, animals were group-housed in individually ventilated cages (IVC; 30 cm × 16 cm × 16 cm), serviced by a central airflow system (Tecniplast, IVC Green Line—GM500), under standard housing conditions (stably controlled 12 h:12 h light/dark cycle, temperature of $23 \pm 2\,°C$, humidity of 55% and sufficient bedding ad nesting material) and were provided with a standard research diet (Altrominute1318, AltrominuteGmbH, Germany) and water *ad libitum* at all times. Two weeks prior to the experimental testing phase, male mice were single housed and female mice were pair-housed. All experiments and protocols were performed in accordance with the European Communities Council Directive 2010/63/EU and were approved by the committee for the Care and Use of Laboratory Animals of the Government of Upper Bavaria. All effort was made to minimise any suffering of the animals throughout the experiments.

### Generation of developmental Fkbp5^Nex and virally induced NexTcf4OE mouse lines

The *Fkbp5^Nex* genetic mouse line was generated by breeding *Fkbp5^lox/lox* mice with Nex-Cre mice[72,43]. This resulted in loss of FKBP51 in glutamatergic neurons of the forebrain (including the neocortex, amygdala, olfactory bulb and hippocampus, but not the dentate gyrus) from embryonic day 11.5 onwards[72]. The specificity of the deletion of FKBP5 in glutamatergic neurons was previously demonstrated[43].

TCF4 overexpression (OE) in glutamatergic neurons of the CA1 (*Nex^Tcf4OE*) was achieved by injecting a cre-dependent AVV-CMV-DIO-mTcf4 virus (Vector Biolabs, Malvern, PA, USA) bilaterally into the CA1 of Nex-Cre female mice. Female mice of the control condition were injected with an AVV-CMV-DIO-eGFP control virus (Vector Biolabs, Malvern, PA, USA) in the same region. Injections were performed via stereotaxic surgeries as described previously[31]. In short, 5-month-old female mice were anaesthetised with isofluorane and fixated in a stereotaxic apparatus. Following preparatory actions, 700 nL of the Tcf4 OE or control viruses were bilaterally injected by using a 33ga blunt tip small hub removable needle (Hamilton, art. No. 7762-06) at a flow rate of 100 ng/min in the CA1. The CA1 was targeted by using the following coordinates: for right injections, 2.2 mm posterior, 2.2 mm lateral, and 1.5 mm ventral from Bregma; for left injections 2.2 mm Posterior, 2.2 mm Lateral, and 1.6 mm ventral from Bregma. Following surgeries, animals received the painkiller meloxicam (2 mg/kg for three days in the drinking water) and were monitored closely up till 7 days post-surgery.

### Limited bedding and nesting material paradigm

In order to investigate the long-term consequences of ELS exposure, mice were exposed to the limited bedding and nesting material (LBN) paradigm that was originally described by Rice and colleagues[73]. Male *Fkbp5^Nex* mice were paired with *Fkbp5^lox/lox* females for breeding purposes. Throughout pregnancy, females were single-housed and monitored daily for the birth of pups. The day of birth of the litter was considered postnatal day 0 (P0), and dams and litter were then assigned to either the ELS or control condition. At P2, dams and pups were checked and put in a fresh cage. If assigned to the control condition, dams and pups were returned to standard housing conditions, with a regular amount of Nestlets (Ancare, Bellmore, NY, USA; 2 full pieces). Dams and pups in the ELS condition were, however, put back

in an IVC with a metal grid, placed on the bottom of the cage, and were only provided a very limited amount of bedding and Nestlets material (half a piece) for a period of 7 days. Litter sizes were matched across experimental conditions. At P9, pups of either condition were weighed, and all dams and pups were put in fresh IVCs with standard housing conditions. At P26, animals were weaned, group-housed, and left undisturbed into adulthood until the start of the experimental procedure.

## Behaviour analyses

Behavioural testing was performed to study anxiety-like behaviour and cognitive functioning in a neutral and stressful environment. The open field (OF) test, elevated plus maze (EPM) test, novel object recognition (NOR) and spatial object recognition (SOR) tests were performed subsequently over a period of 6 days between 8 AM and 1 PM. Tests were performed with a minimal interval of 24 h to allow for recovery and baseline restoration, minimising potential carryover effects from previous tests. The Morris Water Maze (MWM) tasks started 4 days after the SOR. The behaviour of the animals was recorded and later tracked using the advanced video tracking software ANY-maze v.7.15 (Stoelting, Dublin, IE). In case manual tracking was required, this was performed by an experienced observer that was blinded to group allocations.

## Open field

In order to asses anxiety-like behaviours and general locomotor activity, the OF test was performed. In this test, mice could freely explore an OF arena (50 cm×50 cm × 50 cm), made out of grey polyvinyl chloride material for a period of 15 min under dimmed light conditions (30 lux). The total distance travelled in the entire OF arena during the full 15 min was taken as a measure for general locomotor activity. Other parameters that were measured were total distance travelled in metres, time spent in seconds and number of entries into the inner zone (dimensions: 26 cm × 26 cm) of the OF, to assess anxiety-like behaviour. For analyses of anxiety-like behaviour, data was separated in bins of 300 s.

## Elevated plus maze

As an additional measure for anxiety-like behaviour, mice underwent the EPM test. For this, animals were placed on an elevated EPM apparatus that consisted of an elevated (50 cm above the ground) cross maze with two open (30 cm × 5 cm × 0.5 cm) and two closed arms (30 cm × 5 cm × 15 cm). Dimmed light conditions were set to less than 10 lux in the closed arms and approximately 20 lux in the open arms. Mice were located in the centre of the cross maze and were allowed to freely explore the maze for 10 min. Anxiety-like behaviour was measured as the amount of entries into the open arms, time spent in seconds and distance travelled in metres in the open arms. Data was analysed in 300 s time bins. To assess the effects of early life stress exposure specifically, the fold change was calculated by setting the mean value of the non-stressed control group to 1 and calculating the fold change of the individual times or distances travelled in the ELS group.

## Novel object and spatial object recognition

In order to evaluate memory performance in a neutral context we applied two tests, assessing memory function on different domains: the NOR (for recognition memory) and the SOR (for spatial memory). During these tests, the ability of the mice to discriminate between a familiar and unfamiliar object and a familiar and unfamiliar location of objects was evaluated. To this end, two separate objects were built out of black and white Lego© blocks that were unique enough to allow discrimination, but not too distinct that it could create a potential bias based on preference for one specific object. Lego blocks were placed in a square OF arena, and mice were allowed to explore objects or their locations for a period or 15 min. Following an inter-trial interval of 30 min, the type of object or the object location was changed, and animals were placed back into the arena where they could explore the novel objects or their locations during a 5-minute retrieval phase. Exploration of the objects was assessed manually and time spent in seconds exploring the objects was analysed.

## Morris water maze

The MWM is a task that is widely used to assess spatial memory performance in mice under a stressful environment[74]. The MWM was performed in a dimly lit square room with 4 unique spatial cues surrounding the pool, in order to ensure spatial navigation. The pool, that was elevated 110 cm above the floor, had a diameter of 150 cm and a height of 41 cm and was filled with water up to the top, leaving an edge of about 5 cm long. The pool was divided into four quadrants (northwest (NW), northeast (NE), southwest (SW) and southeast (SE) quadrant), and an invisible platform was located in a fixed position in the SW quadrant 0.5 – 1 cm below the water surface. The MWM spatial learning task consisted out of two phases the training phase and the probe trial testing phase. The training phase included 5 consecutive training days for males and 6 consecutive training days for females, in which the mice were placed in randomised starting locations in the opposite quadrant (NE) from the platform location. During the training phase, animals were allowed to find the location of the invisible platform within a 90 s learning trial. Upon finding the platform, mice were taken off the platform immediately. If animals did not find the platform location before the end of the training trial, they were guided towards the platform and left to explore the platform area for 10 s, before being removed. Animals were then quickly dried and returned to their home cage. Each day, mice performed 4 consecutive training trials, with an inter-trial interval of 12 to 16 min. As wild-type female mice displayed difficulties in acquiring the MWM task, subsequent cohorts were tested with shorter inter-trial intervals of 10 minutes. During this phase of the MWM, the time in seconds it took the mice to find the platform location was measured. One day subsequent to the training phase, the probe trial testing phase started. For the testing phase, the platform was removed from the pool, and animals were allowed to explore the pool area for 60 s. During this test, the relative distance travelled in metres in the original platform quadrant (SW) versus the adjacent (NW and SE) and opposite (NE) quadrants was evaluated as a measure of spatial memory performance.

## Magnetic resonance imaging

A horizontal BRUKER Biospec 94/20 animal scanner (Bruker BioSpin, Rheinstetten, Germany), operating at 9.4 Tesla and using a transmit/receive cryo-coil with two coil elements, was used to apply structural magnetic resonance imaging (MRI) as previously described[43]. Animals were sedated using 2.5% isoflurane and stereotactically fixated in a prone position on an MR-compatible animal bed, on top of a warm water silicon pad, where they were held under constant inhalational anaesthesia with isoflurane (1.5–2.5% in pressured air, with a flow of 1.5 l/min). Bepanthen cream (Bayer, Leverkusen, DE) was applied in order to prevent drying of the eyes. Bodily signs, such as body temperature and respiration, were continuously checked and kept in a constant range (body temperature: 36–38 °C; respiration: 80–120 breaths/minute), by either adjusting the temperature of the warm water silicon pad or the depth of isoflurane anaesthesia. For the collection of MR images, first, general adjustments of the system and collection of localiser scans were performed, after which a 3D T2*-weighted image was acquired using a FLASH sequence with TE = 6.25 ms, TR = 34.1 ms, flip angle 10°, matrix size 256 × 166 × 205 points, resolution 0.077 mm isotropic, 2 averages, with fat and outer volume suppression. The acquisition time for the 3D was 41 min 8 s.

## Image processing

Image processing was handled as previously described[43]. The voxel dimensions were artificially increased by a factor of 10, to better match the human brain SPM default values. Optimised brain extraction was based on a three-step procedure, including repeated segmentation using the (modified) Hikishima templates[75], brain mask generation and spatial denoising. The resulting brain-extracted images were co-registered to the Hikishima T2-weighted reference image. The olfactory bulb and the cerebellum were cut out (due to lower signal intensities caused by the geometry of the surface coil). At last, an SPM12 old segmentation step was performed using the GM, WM and inner CSF compartment tissue templates. Resulting tissue probability maps for GM and WM were imported to DARTEL and normalised with isotropic voxel size 0.7 mm to create a study specific template. Normalisation flow fields were transformed into Jacobian deformation fields for later deformation-based morphometry (DBM) analysis[76] and were smoothed with a Gaussian kernel of 4 mm. Both total brain volume (TBV) and volume of the individual tissue compartments were defined from the DARTEL imported images (native space), by summation of the tissue probability values in GM, WM and CSF compartments. The anatomical images were also normalized using the DARTEL flow fields, and a mean image was calculated for later display.

## Tissue collection

At sacrifice, animals were anaesthetised using a lethal dose of isoflurane and subsequent immediate decapitation. Trunk blood was then collected in 1,5 mL EDTA-coated microcentrifuge tubes (Kabe Labortechnik, Nümbrecht-Elsenroth, DE) and saved on ice until further processing. Plasma separation was later achieved by centrifugation (15 min, 8 000 RPM at 4 °C), and samples were stored at − 20 °C. In addition, adrenal glands and brains were extracted. Brains were immediately snap-frozen in isopentane on dry-ice and later stored at − 80 °C. Following collection, adrenal tissue was washed in 9% NaCl, dried and weighed.

## Hippocampal dendritic morphology

In order to further evaluate structural hippocampal consequences of ELS in *Fkbp5Nex* females, the Golgi-Cox staining was applied to visualise and quantify dendritic tree morphology as well as spine density of the dorsal CA1 of 8-month-old female *Fkbp5Nex* and *Fkbp5lox/lox* mice that either underwent the LBN ELS procedure or a control condition. First, mice were sacrificed by a lethal dose of isoflurane, after which transcardial perfusion was conducted using a perfusion pump and ice-cold phosphate-buffered saline (PBS) with 0.1% heparin for approximately five minutes. Following subsequent decapitation, brains were extracted, and the left hemisphere was dissected.

## Golgi-Cox staining procedure

Golgi-Cox staining was then performed with the help of the Bioenno superGolgi Kit (Bioenno Tech, LLC, Santa Ana, CA, USA) according to the manufacturer's protocol. In short, brains were first impregnated for 12 – 14 days in the provided impregnation solution. After rinsing with distilled water, they were then transferred to a post-impregnation buffer for 2 days. Subsequently, CA1 brain sections (150 μm) were collected on a vibratome (HM650V, Thermo Scientific) in a 6% sucrose collection buffer and mounted on gelantine-covered slides (6% gelantine). Following, sections were incubated in the staining solution and incubated in the post-staining solution. After the staining procedure was completed, slides were imaged.

## Imaging and analyses

After the Golgi-cox staining procedure was completed, dorsal CA1 sections were imaged for dendritic length and branching and spine analyses. For dendritic length and branching, Z-stacked images (100 μm stacks) were collected with the Olympus BX61VS slide scanner microscope (Olympus, Hamburg, DE) at 40 x magnification. For dendritic spine analysis, images were made at 100 x magnification, using the Zeiss AXIO Imager M2 with the camera Zeiss Axiocam506 (Zeiss, Oberkochen, DE) and the software Neurolucida (MBF Bioscience, Williston, VT, USA).

A Sholl analysis was performed to determine dendritic branching length with the help of the Simple Neurite Tracer (SNT) plugin from the ImageJ software on 121 collected neurons. The dendritic branch was investigated for a total length of 300um (starting from the soma), and the number of intersections were measured for each 10um section. Dendritic spine analysis was done with the ImageJ (version 1.54) software (for 3 animals in each condition, with 4 – 6 apical and 4 – 6 basal dendritic segments. The length of the individual dendritic segments were measured, spines per segment were counted, and finally, a score per 10 μm of each dendritic segment for each animal was calculated. The average number of spines/10 μm dendritic segments for each condition was used for statistical analyses.

## Electrophysiology

Mice were anaesthetized with isoflurane and immediately decapitated, after which the brain was rapidly removed from the cranial cavity. Subsequently, 350 μm-thick coronal slices of the dorsal hippocampus were collected using a vibratome, in an ice-cold carbogen gas (95% O2/5% CO2)-saturated solution consisting of (in mM): 87 NaCl, 2.5 KCl, 25 NaHCO3, 1.25 NaH2PO4, 0.5 CaCl2, 7 MgCl2, 10 glucose, and 75 sucrose. Brain slices were then incubated in carbogenated physiological saline (containing 125 mM NaCl, 2.5 mM KCl, 25 mM NaHCO3, 1.25 mM NaH2PO4, 2 mM CaCl2, 1 mM MgCl2, and 10 mM glucose) for 30 min at 34 °C, followed by incubation at room temperature (23–25 °C) for at least 1 h. All electrophysiological measurements were conducted at room temperature. Slices assigned to the CORT condition (*Fkbp5Nex* CORT and *Fkbp5lox/lox* CORT) were stored for 1 h in carbogenated physiological saline, containing 1 μM CORT (Sigma-Aldrich Corticosterone, product nr. 27840, dissolved in 0.01 % EtOH; Merck KGaA, Darmstadt, DE). Brain slices in all other conditions (Ctrl and ELS conditions) were pre-incubated with a carbogen physiological saline vehicle solution, containing 0.01% EtOH. Following pre-incubation with CORT or vehicle solution, slices were washed for 30 minutes in pure carbogenated physiological saline. Slices were then transferred to the recording chamber, where they were superfused with carbogenated physiological saline (4-5 ml/min flow rate). Field excitatory postsynaptic potentials (fEPSPs) at CA3 - CA1 synapses were evoked by square-pulse electrical stimuli (50 μs pulse width) delivered via a bipolar tungsten electrode (50 μm pole diameter, ∼ 0.5 MΩ nominal impedance) to the Schaffer collateral-commissural pathway. fEPSPs were recorded using glass microelectrodes (filled with physiological saline, ∼1 MΩ open-tip resistance) that were placed into the CA1 stratum radiatum. Voltage stimulation intensity was adjusted accordingly to produce a fEPSP of ∼ 50% of the amplitude at which a population spike appeared. Recording data were low-pass filtered at 1 kHz and digitised at 5 kHz. Before and after LTP induction, which was induced by high-frequency stimulation (HFS, 100 Hz for 1 s), a single stimulation pulse was delivered every 15 s to the neural tissue.

## RNA sequencing

**RNA extraction.** Hippocampal tissue of 6 mice per condition (female *Fkbp5Nex* and *Fkbp5lox/lox* mice of the ELS and Ctrl condition) was collected from frozen brains via punches, using a 1 mm-diameter punching tool. Both dorsal and ventral hippocampal punches were included. Tissue was then immediately transferred into 1.5 mL DNA LoBind Safelock Eppendorf tubes that were kept on dry ice. Following collection, tissue was again stored at − 80 °C. RNA isolation was then later achieved with the help of the miRNeasy Mini Kit (cat. no. 1038703, QIAGEN, Venlo, NL) RNA extraction kit, according to the manufacturer's protocol.

**RNA sequencing and analyses.** For all steps up until filtering, samples were analysed together with samples form a different experiment. All subsequent analyses were separately conducted for the hippocampus tissue samples of this study. RNA quality control, library preparation, transcriptome sequencing, and RNA sequencing analyses were performed on-site by the company Novogene UK (Novogene Europe, Cambridge, UK) according to their standardised protocols. For cleaning of the data, reads containing adaptor, reads containing poly-N and low-quality reads were removed from the raw data. At the same time, Q20, Q30 and GC content were calculated from the clean data. The index of the reference genome was built using Hisat2 v2.0.5 and paired-end clean reads were aligned to the reference genome using Hisat2 v2.0.5. FeatureCounts v1.5.0-p3 was used to count the reads numbers mapped to each gene. Then, the FPKM of each gene was calculated based on the length of the gene and the read count mapped to this gene. The subsequent analysis was performed in R version 3.6.1. Genes with less than 10x coverage across all samples in each experimental group in each brain region were removed ($n = 4$ experimental groups: Fkbp5$^{lox/lox}$ and control; Fkbp5$^{Nex}$ and control; Fkbp5$^{lox/lox}$ and ELS; Fkbp5$^{Nex}$ and ELS). 16,621 genes were left after this filtering step and 48 samples. To identify outliers, we performed a principal component analysis (PCA). Samples with a distance of more than 2.5 standard deviations from the mean in the first principal component were excluded, which lead to the removal of one sample. Surrogate variable analysis (SVA) was applied to account for unwanted variation in the data.

**Differential expression (DE) analysis.** DE analysis between 4 comparisons (ELS: Fkbp5$^{Nex}$ vs. Fkbp5$^{lox/lox}$; Ctrl: Fkbp5$^{Nex}$ vs. Fkbp5$^{lox/lox}$; $^{Fkbp5Nex}$: ELS vs. Ctrl and Fkbp5$^{lox/lox}$: ELS vs. Control) each brain region was set up. The expression data was normalized and transformed using the vst function of DESeq2 v1.24. We tested for DE with DESeq2, including significant surrogate variables, and reported the genes with a false discovery rate (FDR) below 2% as significant.

**Constructing gene networks and enrichment analyses.** In order to detect underlying pathways containing modules of co-expressed genes, a weighted correlation network (WGCNA) analyses was performed in addition to the differential expression analysis. We used R package WGCNA with a soft threshold of 10, deep split of 4, min. module size of 30 and merge cut height of 0.15. to construct the co-expression network. In addition to this, hub genes of the revealed co-expressed modules or modules were identified. In our analysis, hub genes were identified using the 'chooseTopHubInEachModule' function within the WGCNA package, which isolates the gene in each identified module exhibiting the highest connectivity. Furthermore, transcription factor enrichment analyses for selected WGCNA modules was carried out with the online enrichment analysis tool WEB-based Gene SeT AnaLysis Toolkit (WebGestalt). Using the software Knowing$^{01}$ (Knowing$^{01}$ Gmbh, Munich, Germany), we then further identified which genes of the significant DEGs, WGCNA modules or enriched transcription factors and their regulated genes were enriched in publicly available human GWAS datasets for psychiatric diseases. For this, we selected GWAS datasets of childhood traumatic events in both sexes and in females only (UK Biobank)[77], PTSD in both sexes or in females only (PGC-PTSD Freeze 2 GWAS)[78], MDD[79], that were obtained from the psychiatric genomic consortium website (PCG).

**Reverse transcription and quantitative real-time polymerase chain reaction (qPCR)**

Messenger RNA (mRNA) samples were extracted using the miRNeasy kit according to the manufacturer's instructions (Qiagen). Quantification of mRNA levels was carried out using quantitative real-time PCR (qPCR). Total RNA was reverse transcribed using the iScript cDNA Synthesis Kit (Bio-Rad). Real-time PCR reactions were run in triplicate using the ABI QuantStudio6 Flex Real-Time PCR System and data was collected using the QuantStudio Real-Time PCR software (Thermo Fisher Scientific). Expression levels were calculated using the standard curve absolute quantification method. The endogenous expressed genes *Hprt* and *Polr2b* were used to normalise the data. Used primers are shown in Supplementary Table S2.

## Statistical analyses

Statistical analyses for behavioural, CA1 dendritic spine density and branching and electrophysiological analyses were carried out in R studio (R.4.2.0) or GraphPad Prism 9. Statistical assumptions were checked by using a Shapiro-Wilk test for Normality and a Levene's test for equality of variances. In case these assumptions were violated, non-parametric statistical tests were performed or a boxcox transformation was conducted to normalize the data. Data including 4 groups (*Fkbp5$^{Nex}$* ELS and Ctrl; *Fkbp5$^{lox/lox}$* ELS and Ctrl) were analysed using a two-way ANOVA for ELS exposure by genotype to test for main differences in ELS exposure and genotype or their interaction (ELS x genotype), with post-hoc *t* test. For analysis of field potential recordings and hippocampal dendritic branching, an average value per group was precedingly calculated. For data with 2 groups (*Nex$^{Tcf4OE}$* vs. Ctrl), an independent sample *t* test or non-parametric Mann-Whitney U test was applied to test for group differences. Outliers were identified as values greater than 2 times the standard deviation (SD) from the mean (M) and excluded from analyses. Graphs were constructed with GraphPad Prism 9 or R studio, and part of the figures was created with the help of Biorender.com. *P*-values of less than 0.05 were considered statistically significant, and a statistical trend was recognised for *p*-values of $0.1 \geq P \geq 0.05$.

MRI data was analysed with a two-way ANOVA (ELS x genotype and interaction if statistically significant) with Tukey post-hoc testing for TBV and tissue compartments to detect differences in brain tissue composition and total brain size. An independent 2-factorial model in SPM12 (ELS x genotype) for *Fkbp5$^{Nex}$* animals vs. *Fkbp5$^{lox/lox}$* animals and for ELS vs. control mice, respectively, was used to compare smoothed jacobian deformation fields. Analyses included TBV as a covariate. If not stated otherwise, reported results survive an FWE correction at the cluster level ($p_{FWE,cluster} < 0.05$), with a cluster collection threshold of $p < 0.005$ uncorrected.

### Reporting summary

Further information on research design is available in the Nature Portfolio Reporting Summary linked to this article.

## Data availability

RNAseq data have been deposited in Gene Expression Omnibus under accession number GSE256468. All other data are included in the manuscript are provided in the Supplementary Information/Source Data file. Source data are provided in this paper.

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

## Acknowledgements

The authors thank Daniela Harbich and Bianca Schmid for their excellent technical assistance and support. We also thank Stefani Unkmeir, Sabrina Bauer and the scientific core unit "Genetically Engineered Mouse Models" for genotyping assistance. This work was supported by the SCHM2360/3-1 and SCHM2360/5-1 grants (to M.V.S.) from the German Research Foundation (DFG) and the "Kids2Health" grant of the Federal Ministry of Education and Research (01GL1743C; to M.V.S.).

## Author contributions

L.v.D. and M.V.S. conceived the study and designed the experiments. L.v.D. performed all animal and MRI experiments and was responsible of analyses and/or interpretation of the behavioural and molecular data. A.A., T.S., D.M., S.M., H.Y., R.H., J.B., S.N., D.H., and J.P.L. assisted in the (animal) experiments. T.S. and M.C. assisted in the MRI animal work, analysed the MRI data, and provided scientific advice on the technical aspects and experimental design of the MRI experiments. D.M., M.B., GRammes, and M.E. conducted and analysed electrophysiological measurements. J.M.D. provided scientific expertise for establishing genetic mouse lines. GRehawi and J.K.A. performed analysis of the RNA sequencing data and J.P.L. provided scientific advice for the interpretation of the RNA bulk sequencing data. L.v.D. wrote the initial draft of the manuscript, and M.V.S. supervised the research.

## Funding

## Competing interests

The authors declare no competing interests.
