## [Transparent Peer Review file · Nature Communications]

FKBP51 in glutamatergic forebrain neurons promotes early life stress inoculation in female mice

Corresponding Author: Dr Mathias Schmidt

Version 0:

Reviewer comments:

Reviewer #1

(Remarks to the Author)

This paper describes the effects of early life stress (ES) on anxiety and memory related behavior in male and female mice later in life, in particular in interaction with Fkbp5. It was found that ES in female mice has anxiolytic effects later in life and enhances memory under stressful conditions, effects that are modulated by Fkbp5. Using state-of-the-art approaches the authors also show that Tcf4 plays a potential role in these effects. Such effects are predominantly present in female mice, but less so in male mice.

As such the paper is interesting in showing that ES effects are different in male and female mice and showing some mechanistic evidence.

Major questions:

- 1) Evidence needs to be provided that Fkbp5 is specifically targeted in glutamatergic neurons.
- 2) Is expression of MR and GR altered after ES and in mutant mice?
- 3) It will be important to test downregulation of Tcf4 in Fkbp5lox/lox mice.
- 4) Wildtype mice do not seem to learn in Morris Water Maze. Is that correct? Please explain, since this questions the test.
- 5) The hyperactivity of KO mice needs to be discussed in the context of behavioral results.

Minor questions:

- 1) It will be good to describe the (absence of) effects in male mice in the abstract.
- 2) Does the fact that animals are subsequently tested also affect behavioral testing? Can the authors discuss?

Reviewer #2

(Remarks to the Author)

In this manuscript, Doeselaar et al report the sex-specific effect of early life stress on female mice that either express or lack FKBP51 in excitatory forebrain neurons. They present data to support the idea that FKBP51 expression in excitatory neurons may facilitate adaptive effects of ELS exposure on emotional regulation, cognitive behavior, and neuronal structure and function. They also suggest TCF4 as a potential target that underlies the FKBP51-mediated effects of ELS on brain and behavior. The use of complementary methods, human data, and multiple models are strengths of this paper. Overall, the data are interesting and important, but additional data are needed to fully support the claims that are made.

- (major) While the data supporting the behavioral effect in females is clearly shown, the conclusions made in the subsequent figures are not fully supported since male mice are excluded. Males need to be included in the MRI, neuronal, electrophysiological and RNAseq datasets to understand which changes are specific to females with and without ELS exposure are important for interpretation. It is possible that some of these same changes will be found in males, which will rule them out as contributing to the behavioral effects in this model.
- (major) The behavioral changes in the TCF4 OE mice are modest and are unrelated to ELS changes via FKBP51. The data shown do not justify the claim made in the title of this paper – “via a Tcf4-mediated pathway”. Additional data are needed to support this claim.

- Are there changes to TCF4 regulated genes in the Fkbp5Nex mice?
- The sex of the mice used for electrophysiology is unclear
- Prior studies have described an inverse relationship between TCF4 and FKBP51. Reference to some of these prior studies would be helpful in the discussion.
- The labeling of the figure legends, including the titles, sex, and significance bars should be significantly improved throughout the figures. Some figures are challenging to fully understand in their current state.

Reviewer #3

(Remarks to the Author)

This is an interesting mouse study exploring the combined effect of KO of FKBP51 in forebrain neurons and a limited nesting protocol, to induce ELS, in both male and female mice in adulthood.

The findings show significant differences in the behavioural responses to ELS in WT and KO mice, with a dampening of the stress evoked response that were more marked in females.

The data are well presented, in a logical order, and the descriptions are clear. The authors extend their findings to demonstrate genotype specific changes in brain volume and electrophysiological field potential recordings, then go on to conduct RNA sequencing, identifying a transcription factor TCF4 as a possible regulator.

It is unclear whether changes in maternal behaviour was observed in response to ELS, as well as conditional KO of FKBP1 (not reported). More information is required regarding the number of litters used, number of pups tested per litter, etc. There are some other methodological matters that would be good to get clarification on.

Overall, an interesting paper that will be of interest to those working in the area of early life stress, and sex related behavioural differences.

My other comments are below.

Abstract

Throughout the abstract, the term 'interactive effects' is used several times to indicate response to the combined impact of ELS and KO (brain volume, electrophysiology), such that the directionality of the response cannot be discerned. If space permits, it would be good to have that nuance described.

Introduction

The introduction provides a comprehensive review of the impact of ELS on behaviour, and the role of Fkbp5 based on primate data... It may assist the reader to have clear indication of the species in which the work being cited was done.

From my reading of the paper, it would assist to have a clearer take on how widely a so-called 'pro-resilient' response to ELS has observed in females - across the ELS literature, and across species.

Methods

Line 494

The method of extracting hippocampal tissue provides no detail of the region(s) sampled, number of punches per region, etc.

Figure 1A suggests dorsal hippocampus, but this is never specified.

It is not clear at present, if litter sizes were similar across the genotypes, and across ELS/Control groups - as this might impact maternal care, suckling etc- this information should be provided.

A related question - Was maternal behaviour assessed in any way (eg nursing behaviour, time to retrieve pups etc?).

Re behaviour- There is large variability (as expected) between individuals - so, did you examine for maternal effects (eg in a given litter, was it the case that male and female offspring behaved differently) - as described for the group means ?

The group numbers differ for the males - in Supp Figure 2, the n of the WT control males is 18- more than twice that of some of the other groups - the reason for this is unclear - were multiple groups of mice used? If so, why?

How were outliers considered, and treated.

N is not provided in Figure legends

Were all of the behaviour tests analysed by ANY-maze?.

Was there any features of the way the MWM protocol was implemented that might have contributed to the effect seen at day 3 of the testing phase ?

Results and discussion

The behavioural and phenotype data are easy to follow,

For Figure 2, I think it would assist the reader to have A, B and C indicated/explained in the Figure legend - not all of the abbreviations appear.

FKBP5 interacts with steroid receptors other than the GR, including the MR, progesterone (PR), estrogen (ER), and androgen (AR) receptors – might this underly the effects observed here ? Is there evidence for altered estrogen or testosterone signalling ?

Given that a mechanism by which ELS can alter motivated behavior is through regulation of gonadal hormones (Eck & Bangasser 2020, Neurosc Biobehav Reviews), perhaps there could be more comment regarding this aspect in the discussion, and suggestions for future work to clarify.

Are there agents that block the activity of FKBP5 - it would be good to test these, as an alternate means to inhibit FKBP5.

Towards the end of the discussion the authors comment on human evidence linking TCF4 and neurological/psychiatric disorders, but the comment regarding 'compelling evidence for TCF4-mediated transcriptional regulation, especially in females', driving a pro-resilient phenotype requires some moderation.

Also what do the authors think regarding the cross species relevance of these findings ?

Minor points

line 37 'in' is repeated

line 50 '.....often disabling them to fully participate ...'
should be modified - rather use
'.....often preventing them from fully participating in... ...'
OR '.....often disenabling them to fully participate ...'

Line 87 '.....FKB51 induction and endured circulating glucocorticoid concentrations after stress'.
Replace with
'.....FKB51 induction and prolonged elevation in circulating glucocorticoid concentrations after stress'. Or something similar.

Line 153 - '... were most prominently present in the female /.....'
Could be simplified to '... were most prominent.....'

Line 212 WT is spelt out in full

Line 299 the phrase '...a bulk in research' Could be modified - perhapsa body of research, or some such.

Line 307 '.... a gap in female research ...' change to
'.... a gap in research in females ...'

Line 476 can simply read '...saline, containing 1 uM CORT (Sigma-Aldrich).

Line 518 - off should be 'of'

'principal' is sometimes misspelt (principle incorrectly used)

Version 1:

Reviewer comments:

Reviewer #1

(Remarks to the Author)

The authors have addressed my questions.

Reviewer #2

(Remarks to the Author)

The authors have addressed all of my prior concerns. I have no other suggestions.

Reviewer #3

(Remarks to the Author)

The modified manuscript addresses many of the points raised by reviewers, and provided additional data to back the claims made.

I have some additional comments, as per below.

In the revised abstract:

Line 41-

It would be more helpful to the reader, to give directionality of the changes in volume - note also this could read '..... reflected in volume changes in different cortical regions, the subiculum

Line 93

'....elevation of circulation glucocorticoid concentrations after stress'

Methods

Line 153 change text to '...following ELS exposure, nor were any interactive effects observed'

It is unclear (to this reviewer) how fold change was calculated in Figure 1D.

Line 129- states it was fold change of OA distance vs respective control group - (are these litter comparisons ? or is the comparison simply OA distance (significant) and OA time (not different)- is FC an appropriate term in this case? (if it is the case that these data represent different individuals, the label on Fig 1D needs to be changed). How do the authors interpret the lack of difference in OA time ?

Note line 835,

MWW is used instead of MWM

As pointed out by reviewer 1 previously the data in Figure 1 F are somewhat counter intuitive, as the WT mice do not appear to learn- I believe the text in line 146 regarding this observation should be changed in the results section – in that '..... ELS had a beneficial effect on spatial memory performance in Fkbp5lox/lox mice only ..' is an over simplification, as the ELS mice are simply acting more as would be expected in the test - the caveat of testing conditions in Supp data is not mentioned until later on - so the wording needs to be modified. 'Under the conditions used the female WT did not appear to learn the task as well as the mice exposed to ELS'. See also xxxxxx (the point here is that there is not really a strong improvement in memory by ELS, just a return to what is normally seen under control conditions).

The units for adrenal weight in extended data Fig 1 g/BW need to be corrected for the relevant correction factor used in the calculation.

Extended data Figure 2 - Figure legend typo- 'atency' instead of latency.

Extended data Figure 5 - Why are these data shown as line graphs with a line between the control and ELS condition- this is not appropriate, given they are different mice.

Extended data Figure 7 - corticosteroid receptor is used rather than glucocorticoid receptor, which is used throughout the manuscript. This applies to line 284 of the main text as well.

The description of the data in Extended data Figures 5, 6, 7 (Inventory, as well as the text in manuscript) do not provide information about the region being studied.

Extended data Figure 7 - as above - data are inappropriately shown as line graphs with a line between the control and ELS condition.

Regarding the different sex responses, there are several reviews on the topic, including some recent work. However, there are not many head to head comparisons in both sexes. Conducting male and female cohorts separately makes it difficult to control for the severity of the ELS impact. One article that compared both the male and female rat siblings following maternal separation showed greater behavioural impacts on the males, compared to females- reporting that ... 'female rats were more resilient and had fewer adverse behavioural responses'. Maniam & Morris Psychoneuroendocrinology 2009 "Palatable cafeteria diet ameliorates anxiety and depression-like symptoms

NCOMMS-24-19786-T Rebuttal Letter

Reviewer #1

This paper describes the effects of early life stress (ES) on anxiety and memory related behavior in male and female mice later in life, in particular in interaction with Fkbp5. It was found that ES in female mice has anxiolytic effects later in life and enhances memory under stressful conditions, effects that are modulated by Fkbp5. Using state-of-the-art approaches the authors also show that Tcf4 plays a potential role in these effects. Such effects are predominantly present in female mice, but less so in male mice.

As such the paper is interesting in showing that ES effects are different in male and female mice and showing some mechanistic evidence.

Response: We thank the reviewer for providing their insightful comments and helpful suggestions. Please find our detailed responses below.

Major questions:

1) Evidence needs to be provided that Fkbp5 is specifically targeted in glutamatergic neurons.

Response: We agree with the reviewer that this is a relevant point. In fact, we have previously published data from this specific mouse line and have shown a specific FKBP5 deletion in glutamatergic neurons (van Doeselaar et al., 2023). We have now included this information in the manuscript and specifically refer to the previously published data, validating the utility of the mouse model.

2) Is expression of MR and GR altered after ES and in mutant mice?

Response: The reviewer raises an important question. To address this point, we performed qPCR analysis in both male and female *Fkbp5^{Nex}* mice or wild type littermates, with or without a history of ELS. No significant effects on expression of MR or GR were observed in males or females. We have now included these interesting new data in the manuscript results (Extended Data Figure 7, 8 and 10) and discussed their implications.

3) It will be important to test downregulation of Tcf4 in Fkbp5lox/lox mice.

Response: The reviewer addresses a relevant point, which was not fully clear in the manuscript. The transcription factor TCF4 was selected due to our finding of this factor being the most significant driver of hub genes and psychiatric hit genes related to early life adversity in the dark-orange network, not because it was found to be highly significantly regulated. Nonetheless, we now tested the expression of this gene both in male and female *Fkbp5^{Nex}* mice or wild type littermates, with or without a history of ELS. Confirming the RNAseq results, TCF4 was not significantly changed in expression, regardless of ELS or genotype (Extended Data Figure 7 and 8). It is feasible that the changes of TCF4 expression are too subtle to be detected, or follow a different temporal pattern to be detected under basal conditions. Functional

changes of TCF4 could also be related to posttranslational modifications, which are undetected on the mRNA expression level. We have now included this important point in the discussion.

4) Wildtype mice do not seem to learn in Morris Water Maze. Is that correct? Please explain, since this questions the test.

Response: The reviewer correctly noted that in our initial cohorts, wildtype females performed poorly in the MWM test, whereas females with a history of early life adversity improved their performance over the course of the test. The protocol for the MWM was developed and optimized for male mice, and we therefore considered the possibility that for females this test is slightly more challenging compared to males. We therefore now performed a control experiment in a separate cohort of wildtype C57Bl/6 females, where we reduced the inter-trial interval to 10 minutes. Under these slightly easier conditions, the mice did acquire the task and significantly improved their performance during the training days. These data confirm that the MWM test is valid in females and that the animals can perform it accurately. The more difficult version of the test with longer inter-trial intervals was optimal for testing the cognitive abilities of wild-type and ELS females, as it enabled the detection of improved spatial memory performance in females with a history of ELS. To address the reviewer's concern, we now added the data of our control experiment to the supplemental data of the manuscript (new Extended Data Figure 2D). Furthermore, we pointed out the importance of the inter-trial interval adjustment in female mice in the methods section.

5) The hyperactivity of KO mice needs to be discussed in the context of behavioral results.

Response: We agree with the reviewer that the observed higher activity of female *Fkbp5^{Nex}* mice in the open field is an important finding that warrants further discussion. Specifically, a hyperactivity of mice can affect other behavioural tests. However, in our data set we did not find any indication of a correlation of the increased OF activity of *Fkbp5^{Nex}* animals with measure of anxiety or cognition in the other tests. We can therefore conclude that this phenotype is likely not affecting the interpretation of the behavioural data. We have now included this point in our discussion.

Minor questions:

1) It will be good to describe the (absence of) effects in male mice in the abstract.

Response: In the course of the revisions of this manuscript and also in response to the suggestions of the other reviewers we have now significantly extended the data obtained from male mice. Interestingly, while we observed no clear behavioural phenotype, there were interesting and partially opposite effects observed on structural, functional or genetic levels. We have therefore adjusted our manuscript not only in the abstract, but also the results and discussion section, to reflect these additional data and findings.

2) Does the fact that animals are subsequently tested also affect behavioral testing? Can the authors discuss?

Response: We agree with the reviewer that the order of test can potentially affect subsequent results.

However, this approach is commonly accepted in the research community, as it allows to reduce the required number of animals for multiple behavioural tests. We have ensured an adequate resting period for the animals of a minimum of 24 hours between behavioural tests to minimize the likelihood of carryover effects. We now explicitly mention this in the methods section of the behavioural test.

Reviewer #2

In this manuscript, Doeselaar et al report the sex-specific effect of early life stress on female mice that either express or lack FKBP51 in excitatory forebrain neurons. They present data to support the idea that FKBP51 expression in excitatory neurons may facilitate adaptive effects of ELS exposure on emotional regulation, cognitive behavior, and neuronal structure and function. They also suggest TCF4 as a potential target that underlies the FKBP51-mediated effects of ELS on brain and behavior. The use of complementary methods, human data, and multiple models are strengths of this paper. Overall, the data are interesting and important, but additional data are needed to fully support the claims that are made.

Response: We thank the reviewer for his/her positive and supporting evaluation of our work. We address all raised comments in detail below.

• *(major) While the data supporting the behavioral effect in females is clearly shown, the conclusions made in the subsequent figures are not fully supported since male mice are excluded. Males need to be included in the MRI, neuronal, electrophysiological and RNAseq datasets to understand which changes are specific to females with and without ELS exposure are important for interpretation. It is possible that some of these same changes will be found in males, which will rule them out as contributing to the behavioral effects in this model.*

Response: The reviewer raises a relevant point and we agree that additional data were needed to better understand the sex-specific effects beyond behavioural differences. To address this point, we have now included new experiments and data in male mice. Specifically, we now included data for structural MRI (Extended Data Figure 3), dendritic branching and spine density, electrophysiology (Extended Data Figure 4) and gene expression (Extended Data Figure 8) for male mice. Indeed, these data allowed us to draw more specific conclusions on the differential effects of FKBP5 x ELS interactions in males and females. Interestingly, males showed less pronounced effects of this gene by environment interaction not only on a behavioural level, but also on neuronal structure and when assessing hippocampal gene expression. On the other hand, we did observe a clear genotype effect on hippocampal LTP induction similar to the one observed in females. Furthermore, MRI structural analysis revealed significant structural alterations that are distinct from those observed in females. Collectively, these data further strengthen our conclusion on the striking sex differences in the interaction of FKBP5 with early life adversity and also underscore that these mechanisms need to be studied on all levels of organization. The new data are now integrated in the results section and appropriately discussed.

• *(major) The behavioral changes in the TCF4 OE mice are modest and are unrelated to ELS changes via FKBP51. The data shown do not justify the claim made in the title of this paper – “via a Tcf4-mediated pathway”. Additional data are needed to support this claim.*

Response: We agree with the reviewer that while we provide compelling evidence of an involvement of TCF4 in regulating emotional and cognitive phenotypes, thereby paralleling the observed effects of ELS exposure, the data do not allow to conclude a direct causality. We therefore acknowledge the reviewer's comment that our results do not prove a singular causality, and additional pathways certainly contribute

to the observed phenotypes. In the revised manuscript we have therefore now moderated this claim and adjusted the text in the title, abstract and discussion.

- *Are there changes to TCF4 regulated genes in the Fkbp5^{Nex} mice?*

Response: Under non-stress conditions, TCF4 target genes as Zic1 or Tcf7l2 are not differentially expressed between Fkbp5^{Nex} and wild type litter mates. In females, these genes are increased in their expression following early life stress, while this effect is absent in Fkbp5^{Nex} females. We have now also included qPCR analyses of these TCF4 target genes in male mice, but no significant main or interaction effects were observed (new Extended Data Fig. 8). These data strengthen our conclusion on the female-specific importance of the TCF4-FKBP5 interaction and we have now incorporated these findings in the manuscript.

- *The sex of the mice used for electrophysiology is unclear*

Response: We have now stated the sex of the experimental animals clearly throughout the manuscript.

- *Prior studies have described an inverse relationship between TCF4 and FKBP51. Reference to some of these prior studies would be helpful in the discussion.*

Response: We thank the reviewer to point out this recent work. Indeed, Khan et al recently observed an inverse relationship of TCF4 and FKBP5 expression in skeletal muscle of type 2 diabetes patients (Khan et al., 2023). We now included these findings in our discussion.

- *The labeling of the figure legends, including the titles, sex, and significance bars should be significantly improved throughout the figures. Some figures are challenging to fully understand in their current state.*

Response: We appreciate the reviewer's feedback regarding the labeling of the figure legends. We have carefully reviewed all figure legends and made significant improvements, including the points mentioned by the reviewer.

Reviewer #3

This is an interesting mouse study exploring the combined effect of KO of FKBP51 in forebrain neurons and a limited nesting protocol, to induce ELS, in both male and female mice in adulthood. The findings show significant differences in the behavioural responses to ELS in WT and KO mice, with a dampening of the stress evoked response that were more marked in females. The data are well presented, in a logical order, and the descriptions are clear. The authors extend their findings to demonstrate genotype specific changes in brain volume and electrophysiological field potential recordings, then go on to conduct RNA sequencing, identifying a transcription factor TCF4 as a possible regulator. It is unclear whether changes in maternal behaviour was observed in response to ELS, as well as conditional KO of FKBP1 (not reported). More information is required regarding the number of litters used, number of pups tested per litter, etc. There are some other methodological matters that would be good to get clarification on. Overall, an interesting paper that will be of interest to those working in the area of early life stress, and sex related behavioural differences.

Response: We thank the reviewer for his/her constructive and helpful comments. Please see below our detailed responses, including with regard to the raised issues on missing information and methodology clarifications.

Abstract: Throughout the abstract, the term 'interactive effects' is used several times to indicate response to the combined impact of ELS and KO (brain volume, electrophysiology), such that the directionality of the response cannot be discerned. If space permits, it would be good to have that nuance described.

Response: We have fully revised the abstract and rephrased the summary of the results such that more information regarding the directionality of the observed effects is already included. As noted by the reviewer the space in the abstract is limited, so not all nuances in the data can be fully described here. All details are therefore depicted in the main results text.

Introduction: The introduction provides a comprehensive review of the impact of ELS on behaviour, and the role of Fkbp5 based on primate data... It may assist the reader to have clear indication of the species in which the work being cited was done.

Response: We agree with the reviewer that stating the species where effects are observed is important, and have therefore revised the introduction and discussion to better reflect this information with regard to the cited literature.

From my reading of the paper, it would assist to have a clearer take on how widely a so-called 'pro-resilient' response to ELS has observed in females - across the ELS literature, and across species.

Response: We agree with the reviewer that this is an important point, specifically in relation to sex differences. However, as stated in the introduction and discussion, the available data for females in relation to ELS are sparse, both in human as well as in rodent or primate studies. As a consequence, to the best of our knowledge a pro-resilient consequence of ELS specifically in females has so far not been reported. We now further highlighted this point in the discussion and call for more studies to investigate these effects.

Methods: Line 494. The method of extracting hippocampal tissue provides no detail of the region(s) sampled, number of punches per region, etc. Figure 1A suggests dorsal hippocampus, but this is never specified.

Response: We apologize for this missing information. For RNAseq, we obtained punches of both dorsal and ventral regions of the hippocampus. This information is now included in the methods section.

It is not clear at present, if litter sizes were similar across the genotypes, and across ELS/Control groups - as this might impact maternal care, suckling etc- this information should be provided.

Response: We agree that this is an important information and how specify that litters sizes were matched across experimental groups.

A related question - Was maternal behaviour assessed in any way (eg nursing behaviour, time to retrieve pups etc?).

Response: In the current study we did not assess maternal behaviour, as this might have interfered with the experimental procedure and caused a disturbance of the protocol due to the influence of the observer.

Re behaviour- There is large variability (as expected) between individuals - so, did you examine for maternal effects (eg in a given litter, was it the case that male and female offspring behaved differently) - as described for the group means ?

Response: The reviewer raises an interesting point. We checked for potential litter effects in the behavioural readouts, but did not detect any. The variability in the outcomes can therefore not be directly linked to potential variations in maternal care between litters.

The group numbers differ for the males - in Supp Figure 2, the n of the WT control males is 18- more than twice that of some of the other groups - the reason for this is unclear - were multiple groups of mice used? If so, why?

Response: The animals used for a given test or readout always came from the same cohort of mice and we did not pool data from different batches or experiments. Differences in group numbers arise due to varying breeding success and genotype distribution in litters. Genotyping of the offspring was only performed after weaning, so at the time of ELS the genotype distribution of the animals was still unclear. This resulted in varying group numbers in adulthood and also explains the different n-numbers for the male cohort depicted in Extended Data Figure 2.

How were outliers considered, and treated.

Response: The reviewer may have missed this information in the methods section. As stated there, outliers were identified as values greater than 2 times the standard deviation (SD) from the mean (M) and excluded from analyses.

N is not provided in Figure legends

Response: We agree that n-numbers in the figure legends are highly informative and have included this information now for all figures in the manuscript.

Were all of the behaviour tests analysed by ANY-maze?.

Response: This is correct. We have now included this missing information to the methods section.

Was there any features of the way the MWM protocol was implemented that might have contributed to the effect seen at day 3 of the testing phase?

Response: The MWM protocol was identical for each of the days for acquisition, so any specific effects on testing day 3 are independent of the protocol used.

Results and discussion

For Figure 2, I think it would assist the reader to have A, B and C indicated/explained in the Figure legend - not all of the abbreviations appear.

Response: We agree with the reviewer that the legend for Figure 2 was not fully clear and have now revised it to better reflect the different panels. All abbreviations are now also included.

FKBP5 interacts with steroid receptors other than the GR, including the MR, progesterone (PR), estrogen (ER), and androgen (AR) receptors – might this underly the effects observed here? Is there evidence for altered estrogen or testosterone signalling?

Response: The reviewer raises an interesting hypothesis. Indeed, FKBP51 has been shown to interact with other steroid receptors in addition to GR, and we agree that these interactions might contribute to some of the observed phenotypes. As we did not investigate estrogen or testosterone signalling in the current study, we do not have any evidence for an involvement of these signalling pathways. To address this point, we now included it in the discussion.

Given that a mechanism by which ELS can alter motivated behavior is through regulation of gonadal hormones (Eck & Bangasser 2020, Neurosc Biobehav Reviews), perhaps there could be more comment regarding this aspect in the discussion, and suggestions for future work to clarify.

Response: This point directly relates to the one above, and we fully agree that gonadal hormones may play a critical role in the observed sex differences. We now included a detailed discussion on this point.

Are there agents that block the activity of FKBP5 - it would be good to test these, as an alternate means to inhibit FKBP5.

Response: Indeed, specific inhibitors of FKBP51 have been developed and we and others have used these compounds to study FKBP51 function (Connelly et al., 2020; Gaali et al., 2015; Hartmann et al., 2015; Wedel et al., 2022). However, pharmacological FKBP51 inhibition does not allow to manipulate in a region- and cell-type specific manner, and we have previously shown especially the neuronal cell type can be decisive for FKBP51 function (van Doeselaar et al., 2023). To address this point, we now extended the discussion and added a sentence on the promise of pharmacological FKBP51 inhibition to modulate ELS consequences.

Towards the end of the discussion the authors comment on human evidence linking TCF4 and neurological/psychiatric disorders, but the comment regarding 'compelling evidence for TCF4-mediated transcriptional regulation, especially in females', driving a pro-resilient phenotype requires some moderation.

Response: We agree and have revised this section to moderate the claims.

Also what do the authors think regarding the cross species relevance of these findings?

Response: The reviewer raises a highly relevant discussion point. Rodents share key physiological and genetic features with humans, making them valuable models for studying the stress system and FKBP51, a highly conserved protein across vertebrates. Therefore, the principal findings of our study are likely to be relevant in other vertebrate species, from other rodents to primates and humans. However, there are also substantial differences between our model organism and humans, such as the timing of hormonal fluctuations in females, ecological needs, or cognitive abilities. To fully understand the phenomenon of stress inoculation and related sex differences, it will be indispensable to study these factors in other species, including humans. These findings highlight the potential for translational research to inform our understanding of stress resilience in humans. We have now added this important discussion point to the manuscript.

Minor points

line 37 'in' is repeated

line 50 '.....often disabling them to fully participate ...' should be modified - rather use '.....often preventing them from fully participating in... ..' OR '.....often disabling them to fully participate ...'

Line 87 '.....FKB51 induction and endured circulating glucocorticoid concentrations after stress'. Replace with '.....FKB51 induction and prolonged elevation in circulating glucocorticoid concentrations after stress'. Or something similar.

Line 153 - '... were most prominently present in the female /.....' Could be simplified to '... were most prominent.....'

Line 212 WT is spelt out in full

Line 299 the phrase ‘...a bulk in research ...’ Could be modified - perhapsa body of research, or some such.

Line 307 ‘.... a gap in female research ...’ change to ‘.... a gap in research in females ...’

Line 476 can simply read ‘...saline, containing 1 uM CORT (Sigma-Aldrich).

Line 518 - off should be ‘of’

‘principal’ is sometimes misspelt (principle incorrectly used)

Response: We thank the reviewer for pointing out these language mistakes and inconsistencies. We have corrected all of them.

References:

- Connelly, K.L., Walsh, C.C., Barr, J.L., Bauder, M., Hausch, F., Unterwald, E.M., 2020. Sex differences in the effect of the FKBP5 inhibitor SAFit2 on anxiety and stress-induced reinstatement following cocaine self-administration. *Neurobiol. Stress* 13, 100232. <https://doi.org/10.1016/j.ynstr.2020.100232>
- Gaali, S., Kirschner, A., Cuboni, S., Hartmann, J., Kozany, C., Balsevich, G., Namendorf, C., Fernandez-Vizarra, P., Sippel, C., Zannas, A.S., Draenert, R., Binder, E.B., Almeida, O.F.X., Rühler, G., Uhr, M., Schmidt, M.V., Touma, C., Bracher, A., Hausch, F., 2015. Selective inhibitors of the FK506-binding protein 51 by induced fit. *Nat. Chem. Biol.* 11, 33–37. <https://doi.org/10.1038/nchembio.1699>
- Hartmann, J., Wagner, K.V., Gaali, S., Kirschner, A., Kozany, C., Rühler, G., Dedic, N., Häusl, A.S., Hoeijmakers, L., Westerholz, S., Namendorf, C., Gerlach, T., Uhr, M., Chen, A., Deussing, J.M., Holsboer, F., Hausch, F., Schmidt, M.V., 2015. Pharmacological Inhibition of the Psychiatric Risk Factor FKBP51 Has Anxiolytic Properties. *J. Neurosci. Off. J. Soc. Neurosci.* 35, 9007–9016. <https://doi.org/10.1523/JNEUROSCI.4024-14.2015>
- Khan, S.U., Jannat, S., Shaukat, H., Unab, S., Tanzeela, Akram, M., Khan Khattak, M.N., Soto, M.V., Khan, M.F., Ali, A., Rizvi, S.S.R., 2023. Stress Induced Cortisol Release Depresses The Secretion of Testosterone in Patients With Type 2 Diabetes Mellitus. *Clin. Med. Insights Endocrinol. Diabetes* 16, 11795514221145841. <https://doi.org/10.1177/11795514221145841>
- van Doeselaar, L., Stark, T., Mitra, S., Yang, H., Bordes, J., Stolwijk, L., Engelhardt, C., Kovarova, V., Narayan, S., Brix, L.M., Springer, M., Deussing, J.M., Lopez, J.P., Czisch, M., Schmidt, M.V., 2023. Sex-specific and opposed effects of FKBP51 in glutamatergic and GABAergic neurons: Implications for stress susceptibility and resilience. *Proc. Natl. Acad. Sci. U. S. A.* 120, e2300722120. <https://doi.org/10.1073/pnas.2300722120>
- Wedel, S., Mathoor, P., Rauh, O., Heymann, T., Ciotu, C.I., Fuhrmann, D.C., Fischer, M.J.M., Weigert, A., de Bruin, N., Hausch, F., Geisslinger, G., Sisignano, M., 2022. SAFit2 reduces neuroinflammation and ameliorates nerve injury-induced neuropathic pain. *J. Neuroinflammation* 19, 254. <https://doi.org/10.1186/s12974-022-02615-7>

NCOMMS-24-19786A Rebuttal Letter

Reviewer #1 (Remarks to the Author):

The authors have addressed my questions.

Answer: We thank the reviewer.

Reviewer #2 (Remarks to the Author):

The authors have addressed all of my prior concerns. I have no other suggestions.

Answer: We thank the reviewer.

Reviewer #3 (Remarks to the Author):

The modified manuscript addresses many of the points raised by reviewers, and provided additional data to back the claims made.

Answer: We thank the reviewer.

I have some additional comments, as per below.

In the revised abstract:

Line 41-

It would be more helpful to the reader, to give directionality of the changes in volume - note also this could read

'.... reflected in volume changes in different cortical regions, the subiculum

Answer: The text was adjusted as suggested.

Line 93

'...elevation of circulation glucocorticoid concentrations after stress'

Answer: Changed.

Methods

Line 153 change text to '...following ELS exposure, nor were any interactive effects observed' '

Answer: Changed.

It is unclear (to this reviewer) how fold change was calculated in Figure 1D.

Line 129- states it was fold change of OA distance vs respective control group - (are these litter comparisons ? or is the comparison simply OA distance (significant) and OA time (not different)- is FC an appropriate term in this case? (if it is the case that these data represent different individuals, the label on Fig 1D needs to be changed). How do the authors interpret the lack of difference in OA time?

Answer: To calculate the fold change, the mean value of the non-stressed control group was set to 1 and fold change of the individual times or distances traveled in the ELS group was calculated. This analysis emphasizes the ELS-induced change in behavior respective to the starting baseline of the WT or KO

group, respectively. We have added this additional explanation to the EPM methods section in the supplemental material. Regarding data interpretation, when measuring time in the open arms the data can be more variable due to the fact that anxious animals may enter the start of the open arm and remain in this area without moving much, while less anxious animals will enter the whole open arm and explore it, thus spending a similar amount of time in the open arm area, but covering more distance due to extended exploration. Thus, the parameter time spent in the open arm area can be a bit more variable and the parameter distance traveled in the open arm is generally a more sensitive measure for anxiety-like behavior in this test.

Note line 835,

MWW is used instead of MWM

Answer: Changed.

As pointed out by reviewer 1 previously the data in Figure 1 F are somewhat counter intuitive, as the WT mice do not appear to learn- I believe the text in line 146 regarding this observation should be changed in the results section – in that ‘..... ELS had a beneficial effect on spatial memory performance in Fkbp5lox/lox mice only ..’ is an over simplification, as the ELS mice are simply acting more as would be expected in the test - the caveat of testing conditions in Supp data is not mentioned until later on - so the wording needs to be modified. ‘Under the conditions used the female WT did not appear to learn the task as well as the mice exposed to ELS’. See also xxxxxx (the point here is that there is not really a strong improvement in memory by ELS, just a return to what is normally seen under control conditions).

Answer: We changed the wording of the results text according the reviewer’s suggestion.

The units for adrenal weight in extended data Fig 1 g/BW need to be corrected for the relevant correction factor used in the calculation.

Answer: We thank the reviewer for spotting this mistake. The correct unit is adrenal weight (mg) / body weight (g). This has now been corrected.

Extended data Figure 2 - Figure legend typo- ‘atency’ instead of latency.

Answer: Changed.

Extended data Figure 5 - Why are these data shown as line graphs with a line between the control and ELS condition- this is not appropriate, given they are different mice.

Answer: Line graphs were chosen to illustrate the interactive effects of early life stress and genotype, as they effectively visualize the pattern of change across groups. To clarify, a statement has been added to the figure legends indicating that each data point represents a distinct group of mice, and lines are used solely to enhance visual clarity and highlight the interaction.

Extended data Figure 7 - corticosteroid receptor is used rather than glucocorticoid receptor, which is used throughout the manuscript. This applies to line 284 of the main text as well.

Answer: Changed.

The description of the data in Extended data Figures 5, 6, 7 (Inventory, as well as the text in manuscript) do not provide information about the region being studied.

Answer: This information has now been added in the main text and the figure legends.

Extended data Figure 7 - as above - data are inappropriately shown as line graphs with a line between the control and ELS condition.

Answer: Line graphs were chosen to illustrate the interactive effects of early life stress and genotype, as they effectively visualize the pattern of change across groups. To clarify, a statement has been added to the figure legends indicating that each data point represents a distinct group of mice, and lines are used solely to enhance visual clarity and highlight the interaction.

Regarding the different sex responses, there are several reviews on the topic, including some recent work. However, there are not many head to head comparisons in both sexes. Conducting male and female cohorts separately makes it difficult to control for the severity of the ELS impact. One article that compared both the male and female rat siblings following maternal separation showed greater behavioural impacts on the males, compared to females- reporting that ... 'female rats were more resilient and had fewer adverse behavioural responses'. Maniam & Morris Psychoneuroendocrinology 2009 "Palatable cafeteria diet ameliorates anxiety and depression-like symptoms

Answer: We appreciate the reviewer's point regarding the benefits of direct sex comparisons in ELS studies. While a combined-sex design with head-to-head comparisons would be ideal for controlling ELS severity, such an approach was not feasible in our study due to its complexity. Specifically, in our study we included not only ELS but also genotype in our analyses, and there was an extensive and time-consuming behavioral testing battery required for each animal. To mitigate potential differences in ELS impact between sexes, we carefully matched all experimental conditions across sexes. Furthermore, our separate-cohort design allows for a more nuanced investigation of sex-specific effects of ELS and genotype, which may have been obscured in a combined analysis. We have expanded the discussion to address these points and included the suggested reference.